# RNG2 tethers the conoid to the apical polar ring in *Toxoplasma gondii* to enable parasite motility and invasion

Romuald Haase[1☉], Bingjian Ren[1☉], Albert Tell i Puig[1], Alessandro Bonavoglia[1], Jean-Baptiste Marq[1], Rémy Visentin[1], Nicolas Dos Santos Pacheco[1¤], Bohumil Maco[1], Ricardo Mondragón-Flores[2], Oscar Vadas [1]*, Dominique Soldati-Favre [1]*

1 Department of Microbiology and Molecular Medicine, Faculty of Medicine, University of Geneva, Geneva, Switzerland, 2 Department of Biochemistry, Center for Research and Advanced Studies of the National Polytechnic Institute, Mexico City, Mexico

☉ These authors contributed equally to this work.
¤ Current address: Department of Biochemistry, University of Cambridge, Cambridge, United Kingdom
* dominique.soldati-favre@unige.ch (DS-F); oscar.vadas@unige.ch (OV)

## Abstract

The conoid is a dynamic, tubulin-based structure conserved across the Apicomplexa that undergoes extrusion during egress, gliding motility, and invasion in *Toxoplasma gondii*. This organelle traverses the apical polar ring (APR) in response to calcium waves and plays a critical role in controlling parasite motility. While the actomyosin-dependent extrusion of the conoid is beginning to be elucidated, the mechanism by which it remains apically anchored to the APR is still unclear. RNG2, a protein localized to both the conoid and the APR, has emerged as a strong candidate for mediating this connection. Biochemical analysis revealed that RNG2 is an unstable protein, undergoing extensive proteolytic cleavage both in the parasite and in heterologous expression systems. Its biochemical properties, with the presence of large coiled-coil domains, likely facilitate the formation of concatenated assemblies, enabling RNG2 to serve as a dynamic and resilient bridge between the conoid and the APR. Using a combination of iterative ultrastructure expansion microscopy and immunoelectron microscopy, we confirmed the localization of RNG2 to the 22 tethering elements bridging the APR and the conoid. Conditional depletion of RNG2 led to the striking detachment of the intact conoid organelle from the APR, supporting an essential role for RNG2 as a tether. Cryo-electron tomography of conoid-less parasites revealed that, in the absence of RNG2, the apical vesicle remains anchored to the plasma membrane, while the rhoptries follow the detached conoid. Although RNG2 depletion only mildly reduces microneme secretion, the parasites are immotile and exhibit impaired rhoptry discharge, highlighting the critical role of proper conoid anchorage in motility and host cell invasion. Comprehensive mutagenesis of RNG2 identified distinct regions responsible for binding to the conoid and the APR, and demonstrated

**Data availability statement:** All relevant data are within the paper and its Supporting information files.

**Funding:** The project is funded by the Swiss NSF to D.S.-F. (310030_215445 and CRSII5_198545) https://www.snf.ch/en. A.T.I.P. is supported by a scholarship of the institute of Genetics and Genomics in Geneva - iGE3 https://www.ige3.unige.ch/. B.R. was supported by EMBO Postdoctoral fellowship (ALTF 531-2021) https://www.embo.org/funding/fellowships-grants-and-career-support/post-doctoral-fellowships/. The funders had no role in study design, data collection and analysis, decision to publish, or preparation of the manuscript.

**Competing interests:** The authors have declared that no competing interests exist.

**Abbreviations:** APR, apical polar ring; AV, apical vesicle; BF, binding fiber; CD, cytochalasin D; Cryo-ET, cryo-electron tomography; DMEM, Dulbecco's modified Eagle's medium; IAA, indole-3-acetic acid; ICMT, intraconoidal microtubule; IEM, immunoelectron microscopy; IFA, Indirect immunofluorescence assay; IMC, inner membrane complex; iU-ExM, iterative ultrastructure expansion microscopy; KinA, KinesinA; MS, mass spectrometry; MV, microtubule-associated vesicle; MyoH, myosin H; MyoA, myosin A; PCR, preconoidal ring; PVM, parasitophorous vacuole membrane; SEC, size exclusion chromatography; SPMT, subpellicular microtubule, WB, western blotting.

that the full-length, intact protein is essential for bridging these two structures and for its functional activity. Altogether, RNG2 emerges as a pivotal protein that ensures conoid functionality and coordination in Coccidia.

## Introduction

The phylum Apicomplexa comprises single-cell eukaryotic parasites of high medical relevance, most notably *Plasmodium* spp., *Cryptosporidium* spp., and *Toxoplasma gondii* [1]. Members of this phylum share an elaborate apical complex composed of specialized secretory organelles called rhoptries and micronemes, as well as tubulin-based cytoskeletal elements [2,3]. The apical complex critically coordinates organelle secretion and activation of the actomyosin system to drive gliding motility, invasion, and egress from infected cells [4,5].

The shape of *T. gondii* is determined by scaffolding components, including the inner membrane complex (IMC), the alveolin network, and the subpellicular microtubules (SPMTs), which collectively provide rigidity to the parasite [6,7] (Fig 1). The 22 SPMTs extend approximately two-thirds of the parasite length and emerge from a protein-based structure called the apical polar ring (APR) [8,9] (Fig 1). The conoid consists of a cone of spiraling tubulin fibers [10], surmounted by two preconoidal rings (PCRs) serving as hubs for the assembly of the gliding machinery [11]. The inside of the conoid hosts two short intraconoidal microtubules (ICMTs) [10] with aligned microtubule-associated vesicles (MVs) predicted to participate in rhoptry discharge [12–14] (Fig 1). Importantly, the conoid is a dynamic organelle that extrudes and retracts through the APR, in response to changes in intracellular calcium levels [15,16]. Conoid extrusion relies on F-actin produced by formin 1 positioned at the PCRs, and is powered by myosin H (MyoH), anchored to the cone [11]. The dynamics of the conoid serve as a gatekeeper for the entry of F-actin in the pellicular space, between the IMC and plasma membrane, to reach other glideosome components, such as myosin A (MyoA), and sustain parasite forward motion [11,17].

The APR serves as a microtubule-organizing center for the SPMTs [8,9]. Several mutants that lead to the loss of APR result in disorganization of the SPMTs and disappearance of the conoid [7,18–20]. A few APR proteins have been identified to date in *T. gondii*. KinesinA (KinA) is an early marker of APR, whereas APR1 and RNG1 appear later during daughter cell formation [19,21]. Individual protein downregulation has a minimal effect on parasite replication; however, simultaneous depletion of KinA and APR1 resulted in the loss of the APR, disorganized SPMTs, and drastic defects in gliding motility and invasion [19]. A recent study demonstrated that downregulation of APR2 destabilizes the multilayered structure of the APR, severely disrupting the apico-basal F-actin flux, leading to defects in parasite motility and host cell invasion [22].

RNG2 exhibits an atypical localization, with its carboxy-terminus anchored at the APR and its amino-terminus extending to the conoid [23]. Although this previous study linked RNG2 depletion to signaling defects that impair microneme secretion and invasion, our work provides key mechanistic insights into its structural role in maintaining conoid integrity.

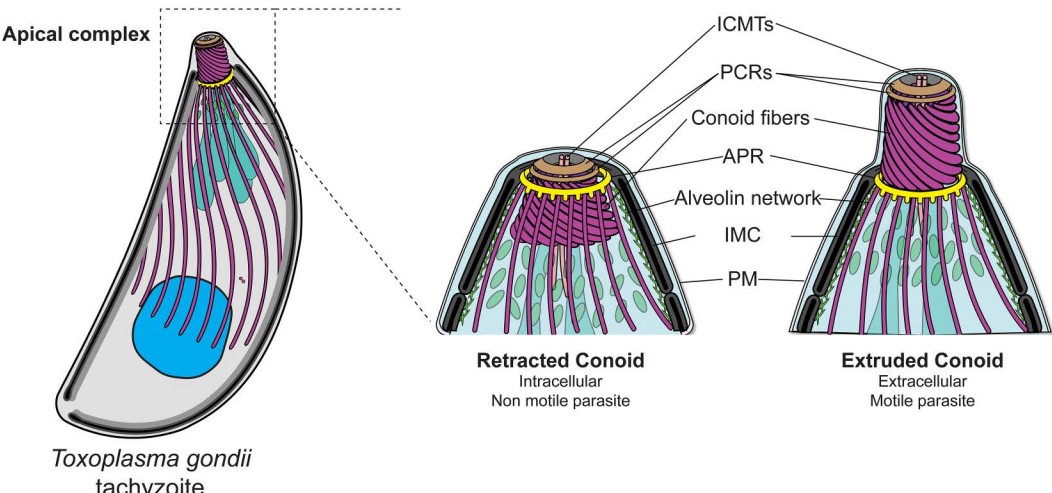

**Fig 1.** *Toxoplasma gondii* **tachyzoite cell organization. Left panel:** general cell organization of *T. gondii* tachyzoites. **Right panel:** Zoomed view of the detailed apical complex. ICMTs: Intraconoidal microtubules; PCRs: Preconoidal rings; APR: Apical Polar Ring; IMC: Inner Membrane Complex; PM: Plasma membrane; Scheme modified from [2].

Here we show that recombinantly produced RNG2 is an unstable protein, exhibiting extensive proteolytic cleavages and oligomerization. This protein, largely composed of coiled-coil regions, can span the distance between the conoid and the APR, serving as a flexible tether that accommodates rapid changes in distance during conoid extrusion and retraction. Iterative U-ExM (iU-ExM) and immunoelectron microscopy (IEM) with polyclonal anti-RNG2 antibodies confirm that RNG2 physically bridges the gap between these two structures. Notably, this is reminiscent of the 22 binding fibers (BFs) previously visualized by EM and proposed to contribute conoid extrusion/retraction [24]. Strikingly, conditional depletion of RNG2 leads to dramatic conoid detachment from the apical pole, reinforcing its essential role in maintaining the APR-conoid connection to ensure proper organelle positioning during retraction.

## Results

### RNG2 is prone to proteolysis and forms oligomers in vitro

*RNG2* codes for a 290 kDa protein that encompasses a central part annotated as tropomyosin domain (Tropo) within a long coil-coiled region flanked by disordered N- and C-termini (Fig 2A). To decipher the biochemical properties of RNG2, recombinant full-length and truncated RNG2 variants with an N-terminal His-Tag and a C-terminal TwinStrep-Tag were expressed in Sf9 insect cells (Fig 2A). All RNG2 constructs showed robust expression; however, only the ΔN+CTer variant exhibited good solubility (Fig 2B and 2C). Full-length RNG2 was completely insoluble, whereas the ΔNTer, ΔTropo, and ΔCTer variants showed partial solubility (Fig 2B and 2C). Western blot analysis with anti-His and anti-Strep antibodies further highlighted the presence of both N- and C-terminally truncated fragments for almost all the variants (Fig 2C). The ΔNTer and ΔN+CTer variants did not display any dominant C-terminally truncated fragment, with a single most-intense band detected with anti-His antibody corresponding to the predicted size of the full-length proteins (Fig 2C).

Purification via the N-terminal His-Tag was not achievable for any of the constructs. In contrast, purification by pulling on the C-terminal TwinStrep-Tag proved successful for RNG2 constructs where the C-terminal 310 residues were truncated (ΔCTer and ΔN+CTer). However, purification required to use salt concentrations largely exceeding physiological conditions (800 mM NaCl). Affinity-purified proteins were subjected to size exclusion chromatography (SEC), and the fractions were analyzed by SDS–PAGE, mass spectrometry (MS), and western blotting (WB). For both C-terminally

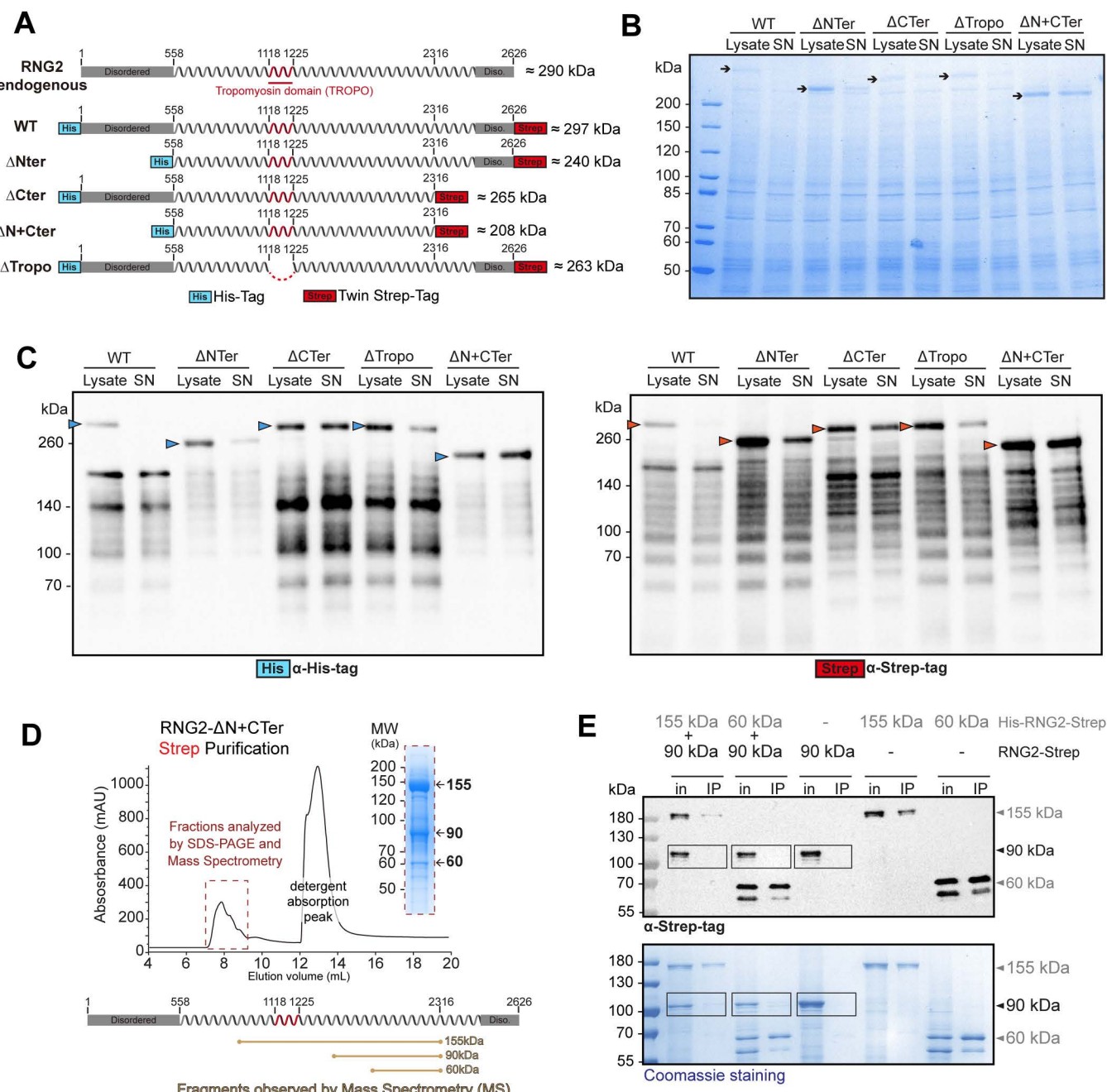

**Fig 2. Biochemical investigation of wild-type and processed RNG2. (A)** Representation of the RNG2 constructs expressed in Sf9 insect cells. **(B)** Insect cell protein expression and solubility of the RNG2 truncation variants, as assessed by Coomassie blue staining. **(C)** Protein expression and solubility of RNG2 truncation variants were assessed via western blotting. **(D)** Size-exclusion chromatography (SEC) chromatogram of the affinity-purified RNG2-ΔN+CTer variant obtained via Strep-tactin purification. A Coomassie-stained gel of the protein-containing fractions of the SEC is shown on the right. The data underlying this figure can be found in S2 Data. **(E)** Co-purification assays using metal-affinity chromatography of cells co-infected with RNG2 fragments assessed by western blot analysis and Coomassie blue staining. All fragments have a C-terminal STREP-tag.

truncated constructs (ΔCTer and ΔN+CTer), purification led to the isolation of the same three most-abundant products with sizes corresponding to 155, 90, and 60 kDa (Figs 2D and S1A). All the fragments coeluted in the void volume of the SEC, corresponding to molecule sizes greater than 600 kDa, indicating that all soluble fragments purified formed large oligomers (Figs 2D and S1A). WB analysis confirmed that the purified products had lost the N-terminal His-Tag (S1B Fig). This provides an explanation for the failure to isolate RNG2 fragments by using the N-terminal His-tag and strongly suggests that these protein constructs are susceptible to proteolytic cleavage. MS analysis confirmed that the purified material corresponded to RNG2 fragments and allowed mapping of the fragment boundaries (Figs 2D and S1C).

To isolate homogeneous RNG2 truncated proteins, each of the 155, 90, and 60 kDa fragments (encompassing residues 985–2290, 1529–2290, and 1793–2290, respectively) were individually expressed and purified via the same workflow as described earlier. Like the larger constructs, each fragment eluted in the void volume by SEC, indicating oligomerization (S1D Fig). To test whether fragments of different sizes could form hetero-oligomers, pull-down experiments were performed on cells co-infected with pairs of fragments of different sizes, where one of the fragments was affinity-purified via a unique N-terminal His-tag (Fig 2E). Purification of either the 155 kDa or the 60 kDa fragment did not result in co-elution with the 90 kDa fragment, indicating that the oligomers observed are likely homo-oligomers. Since homo-oligomerization is thermodynamically favorable [25], the fragments that co-eluted during the SEC of purified ΔCTer and ΔN+CTer constructs likely did not assemble into hetero-oligomers (Fig 2D and 2E).

Collectively, heterologous expression of RNG2 provided evidence that the protein is highly unstable and susceptible to rapid proteolytic degradation. Only a mixture of truncated RNG2 products of the same size could be purified, which were associated with large homo-oligomers containing fragments of 155, 90, and 60 kDa.

## RNG2 decorates 22 tethers between the conoid and the APR

Double epitope tagging at both extremities of RNG2 previously revealed that the N-terminal tag stains the conoid, whereas the C-terminal tag labels the APR [23]. To resolve this conundrum and scrutinize the biochemical properties of RNG2 in *T. gondii*, we introduced epitope tags at the N-terminus (Myc), center (Ty), and C-terminus (HA), along with a C-terminal mAID cassette (Fig 3A). Prediction of the maximum length of RNG2 suggested that its coiled-coil regions could span approximately 215 nm, consistent with the distance between the conoid and the APR (Fig 3A). The endogenous triple epitope-tagged RNG2 clonal line was confirmed by genomic PCR (S2A Fig) and WB analysis (Fig 3B). Indirect immunofluorescence assay (IFA) using the three antibodies (Myc, Ty, HA) confirmed the apical localization of the RNG2 as well as tight regulation under auxin (indole-3-acetic acid [IAA]) treatment (S2B Fig). Several RNG2 truncated products of high molecular weight were detected with anti-HA and anti-Ty antibodies. The specificity of those bands was confirmed by their disappearance in parasites treated with IAA. However, the anti-Myc antibody detected only the full-length version at a high molecular weight with additional smaller fragments around 100 kDa, which is consistent with N-terminal processing events (Fig 3B). In addition, the use of RNG2 rabbit polyclonal antibodies raised against the recombinant central fragments of RNG2 (155, 90, 60 kDa) (Fig 2D), revealed the same processing events (Fig 3B). By comparing the WB obtained from insect cells expressed variants and the ones using all the epitopes-tags in *T. gondii,* using the sizes of the processed products the map we could determine that RNG2 displays five main cleavage sites that are conserved between insect cells and the parasite (S3A Fig). Moreover, in *T. gondii* neither the full length nor processed products could be observed soluble using the three epitope tags (Myc-Ty-Ha) (S3B Fig). Interestingly, the extensive proteolytic cleavage of RNG2 observed in the parasite did not differ between intracellular and extracellular states, indicating that it is constitutive and not dependent on conoid dynamics (Fig 3C). To refine the dynamic behavior of RNG2, the position of the different tags was assessed via U-ExM on extracellular tachyzoites on extruded or retracted conoids (Fig 3D). The N-terminus of RNG2 (Myc) was found at the base of the conoid both in extruded and retracted states, whereas the C-terminus (HA) was always localized on top of the SPMTs at the APR level, confirming the previous observation [23]. The central Ty-tag localized between the conoid and the SPMTs, as did the anti-RNG2 antibodies raised against the central region of RNG2.

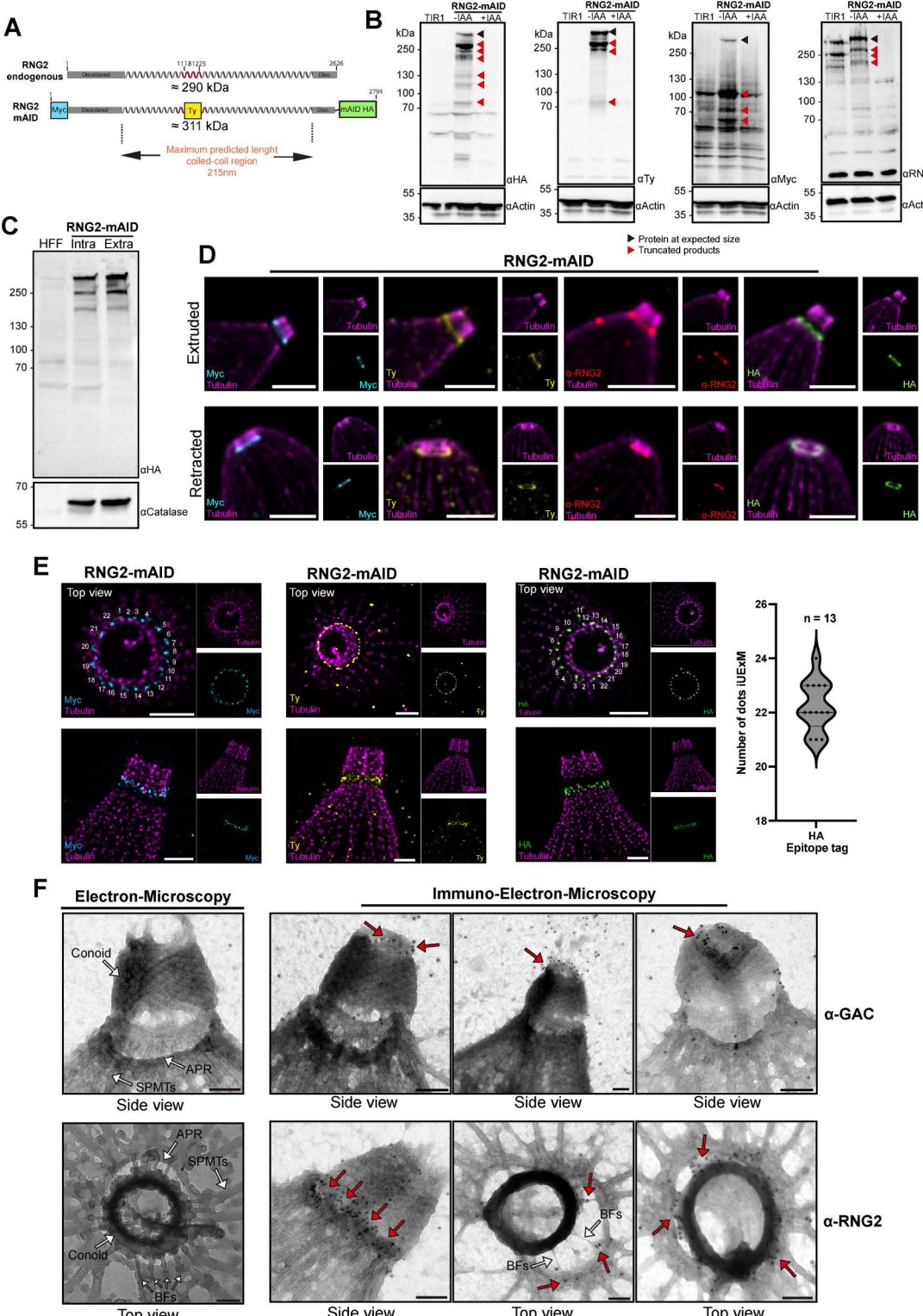

**Fig 3. RNG2 spans between the APR and the conoid, forming 22 tethers. (A)** Illustration of endogenous tagging of RNG2, including three epitope tags and a minimal auxin inducible (mAID) motif. The estimation of the protein length corresponding to the distance coverage of the central part of the protein in linear conformation. **(B)** Protein depletion after 48 h of auxin (IAA) treatment assessed by western blot. **(C)** Western blot of parasite lysate

collected in intracellular or extracellular state **(D)** U-ExM image of the locations of RNG2-mAID depending on the tag insertion and RNG2 polyclonal antibody (recognizing the central region). Images were taken with extruded and retracted conoids. Scale bar = 3 μm. **(E)** Iterative expansion microscopy (iU-ExM) of the N-terminal (Myc), middle (Ty), and C-terminal (HA) tags of RNG2-mAID. Scale bar = 3 μm. Right panel: Distribution of the numbers of dots counted for the anti-HA staining across 13 images. The data underlying this figure can be found in S2 Data. **(F) Left panel:** Electron microscopy images of the *Toxoplasma gondii* apical complex as an ultrastructure control. **Right panel:** Immunoelectron microscopy images obtained with the α-GAC and α-RNG2 antibody. Red arrows = gold particles. Scale bar = 100 nm.

Colocalization of the Ty- and anti-RNG2 tags with HA- and Myc-tags confirmed that RNG2 is oriented with its N-terminal region at the conoid, its central region spanning the space between the conoid and the APR, and its C-terminal region at the APR (S3C Fig). Notably, Ty-Tag colocalized with the N-terminal Myc-Tag only when the conoid was retracted, suggesting a shortening of the protein length in the retracted state (S3D Fig).

To achieve higher resolution, we employed iterative ultrastructure expansion microscopy (iU-ExM), which enables a 16-fold expansion compared with the 4-fold expansion achieved by conventional U-ExM [26]. With increased resolution analysis based on a dozen of images, we observed that the RNG2 C-terminus exhibited approximately 22 puncta aligning with the tips of the SPMTs (Fig 3E). Additionally, around 22 puncta could sometimes be detected at the RNG2 N-terminus on the conoid, although these were less distinct. (Fig 3E).

A study based on electron microscopy (EM) ultrastructural analysis revealed the existence of BFs organized in a cartwheel-like conformation and linking the conoid to the SPMTs [24]. To test whether RNG2 could be a component of BFs, we performed IEM using rabbit anti-RNG2 antibodies (Fig 3F). Although BFs were readily detected in the control samples (Fig 3F), when processed for IEM, many of the BFs disappeared, likely due to the numerous washes and incubations with the antibodies. However, a large accumulation of gold labels was found in the space between the conoid and APR and on the BFs still visible in the preparation (Fig 3F). As a control, we used rabbit polyclonal anti-GAC antibodies, which showed accumulation of gold particles at the PCRs where GAC is localized [11] (Fig 3F). Taken together, the iU-ExM and IEM data showing the spread of RNG2 between the APR and the conoid support its identification as a component of the 22 BFs.

## RNG2 is essential for maintaining conoid attachment to the APR during retraction

Given biochemical and imaging evidence for a role of RNG2 in conoid-APR attachment, we revisited its function, taking advantage of the fast depletion of the protein with the auxin degron system. RNG2 depletion by IAA, as well as deletion of the gene Δ*rng2*, were confirmed to be fitness-conferring by plaque assays (S4A Fig). In intracellular parasites, no obvious cytoskeletal difference was observed in RNG2-depleted parasites under U-ExM (Fig 4A). Conoid extrusion can be triggered by artificial elevation of cGMP levels with BIPPO, a phosphodiesterase inhibitor [27], and unambiguously observed by U-ExM and EM [11,27,28]. Strikingly, in BIPPO-stimulated extracellular RNG2-depleted parasites, the conoid detached from the cell apex and was observed floating in the cytosol by both U-ExM and EM (Fig 4B and 4C). Importantly, conoid detachment was prominently observed in more than 80% of BIPPO-stimulated extracellular parasites. To determine whether actomyosin-driven conoid extrusion triggers conoid detachment, we performed a conoid extrusion assay using parasites pretreated with cytochalasin D (CD), an inhibitor of actin polymerization [11]. Under normal conditions, BIPPO stimulation resulted in more than 84% of parasites exhibiting conoid extrusion, whereas treatment with CD nearly abolished extrusion, with 95% of parasites retaining a retracted conoid (Fig 4D). In contrast, only 9% of BIPPO-stimulated RNG2-depleted parasites showed conoid extrusion, while 77% exhibited conoid detachment. Notably, CD treatment of RNG2-depleted parasites preserved a retracted conoid phenotype, similar to the parental strain (Fig 4D). These results indicate that conoid detachment caused by RNG2 depletion occurs only upon stimulation. EM imaging revealed that the detached conoid remained intact, with PCRs and ICMTs still associated with the tubulin cone (Fig 4C). Additionally, the electron-dense structure capping the 22 SPMTs known as the APR remained unaffected under EM imaging (Fig 4C).

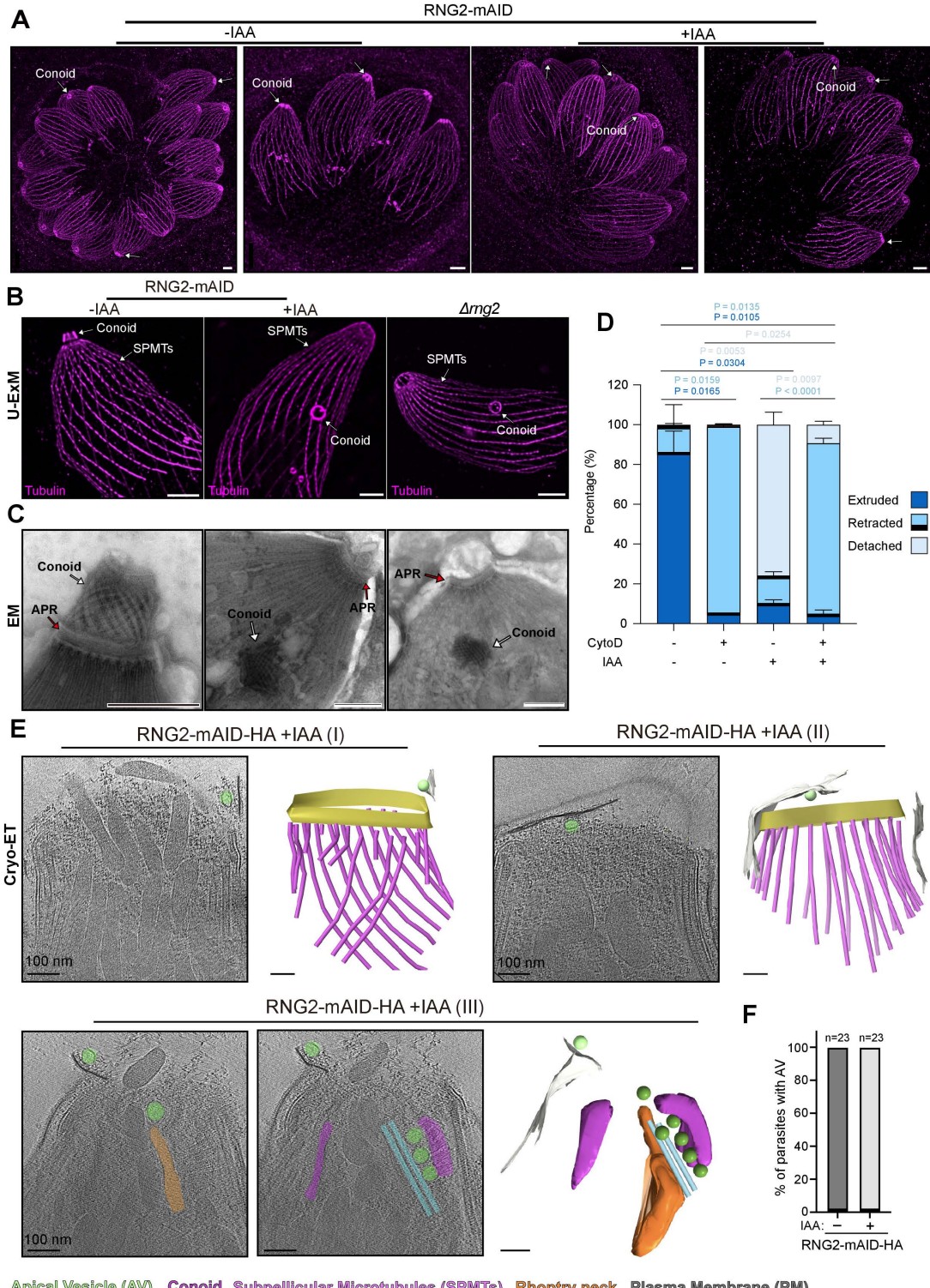

**Apical Vesicle (AV)** **Conoid** **Subpellicular Microtubules (SPMTs)** **Rhoptry neck** **Plasma Membrane (PM)**
**Microtubules associated Vesicles (MVs)** **Intraconoidal Microtubules (ICMTs)** **Apical Polar Ring (APR)**

**Fig 4. RNG2 depletion leads to conoid detachment from the apical pole of the parasite. (A)** U-ExM images of the tubulin cytoskeleton of RNG2-mAID parasites treated for 30 h with auxin showing no cytoskeletal defects. White arrows: conoids **(B)** U-ExM images showing extruded conoid and

detached conoid in RNG2-mAID-depleted parasites and RNG2 knockout parasites (Δ*rng2*). Scale bar = 3 µm. **(C)** Electron microscopy images of negatively stained tachyzoites with extruded conoids and detached conoids in RNG2-mAID-depleted parasites and Δ*rng2* parasites. Scale bar = 500 nm. **(D)** Quantification of the conoid state in three categories: extruded, retracted, and detached. Under all conditions, extracellular tachyzoites were triggered for conoid extrusion via BIPPO in the absence or presence of auxin (IAA) and with or without cytochalasin D (CD). *N* = 100 parasites in 3 biological replicates. The data underlying this figure can be found in S2 Data. **(E)**: Cryo-electron tomography performed on RNG2-mAID parasites with detached conoid. Upper panel shows two parasites with completely detached conoids. Lower panel shows a conoid detached and falling towards the cytoplasm. **(F)**: Quantification of the presence of the apical vesicle (AV) in the tomograms of parasites depleted or not for RNG2. *n* = 23. The data underlying this figure can be found in S2 Data.

To gain further insight into the consequences of RNG2 depletion, cryo-electron tomography (Cryo-ET) was performed on parasites with detached conoids (Fig 4E). While all components of the apical complex were correctly positioned in untreated parasites, the apical vesicle (AV), a plasma membrane-embedded structure essential for rhoptry discharge and invasion, was the only structure that remained correctly positioned in conoid-less parasites (Figs 4E, 4F, S5A and S5B and S1–S3 Movies). Interestingly, we captured the apical end of a parasite with a falling conoid, showing that conoid-associated components such as ICMTs, MVs, and rhoptry necks moved with the conoid while remaining correctly positioned within it (Fig 4E and S3 Movie). In addition, consistent with the positioning of the AV, no difference in localization was observed for the Nd6 protein previously localized at the rosette [29] (S5C and S5D Fig). Taken together, these findings demonstrate that RNG2 is essential for retaining the conoid at the apical pole after extrusion.

### Conoid detachment in the absence of RNG2 disrupts motility and rhoptry positioning

To investigate quantitatively and at the molecular level the consequences of the conoid detachment on the structures of the apical complex, markers of the conoid (CPH1), the PCRs (Pcr4), and the APR (APR1) were epitope-tagged in the background of the RNG2-mAID strain [11,19,30] (Fig 5A). In parasites detaching their conoid, the three components remained correctly positioned, suggesting no alteration of the structures but simply a disconnection (Fig 5A).

*T. gondii* tachyzoites possess eight to 12 rhoptries, with two of them having their necks docked inside the conoid and primed for discharge [13]. Micronemes similarly accumulate at the apical pole of the parasite [31]. Since the conoid is assumed to serve as funnel for the secretion of micronemes and rhoptries, we assessed the impact of conoid detachment on these processes. U-ExM revealed no striking disorganization of the micronemes as they could accumulate apically; however some micronemes were still associated to the detached conoid, pointing to the existence of physical interactions between the conoid and these organelles (Fig 5B). More strikingly, the rhoptries were considerably disorganized in RNG2-depleted parasites (Fig 5B). Consistent with these observations, induced microneme secretion showed only a partial reduction of exocytosis following conditional depletion of RNG2 (Figs 5C and S4B). The level of micronemes secreted allowed RNG2-depleted parasites to successfully egress, lysing both the parasitophorous vacuole membrane (PVM) and host cell plasma membrane upon BIPPO stimulation (Figs 5D and S4C). However, RNG2-depleted parasites remained clustered at the site of host cell lysis, indicating an impairment in motility. This was confirmed by a severe defect in the gliding trails assay (S4D Fig). Because proper rhoptry positioning is crucial for discharge, we assessed rhoptry secretion in RNG2-deficient parasites. Rhoptry secretion was measured indirectly, by monitoring the phosphorylation of host cell nuclear STAT6, a marker for rhoptry discharge [32]. In aspartyl protease 3-depleted parasites, used here as a control, rhoptry secretion is completely blocked [33], whereas RNG2-depleted parasites show a ~50% reduction in rhoptry discharge, comparable to that observed in the Δ*rng2* mutant (Fig 5E). Consequently, depletion of RNG2 resulted in a reduction of invasion efficiency to less than half of that seen in the parental strain (S4E Fig).

To conclude, the disconnection of the conoid from the APR led to detachment of rhoptries and a small portion of microneme from the apical tip, resulting in severe defects of rhoptry discharge as well as a modest defect of microneme secretion. While the invasion capacity of parasite is adversely affected by the failure of rhoptry discharge, the minor effect of microneme secretion does not impair the rupture of the PVM or the host cell. In RNG2-depleted parasites, conoid

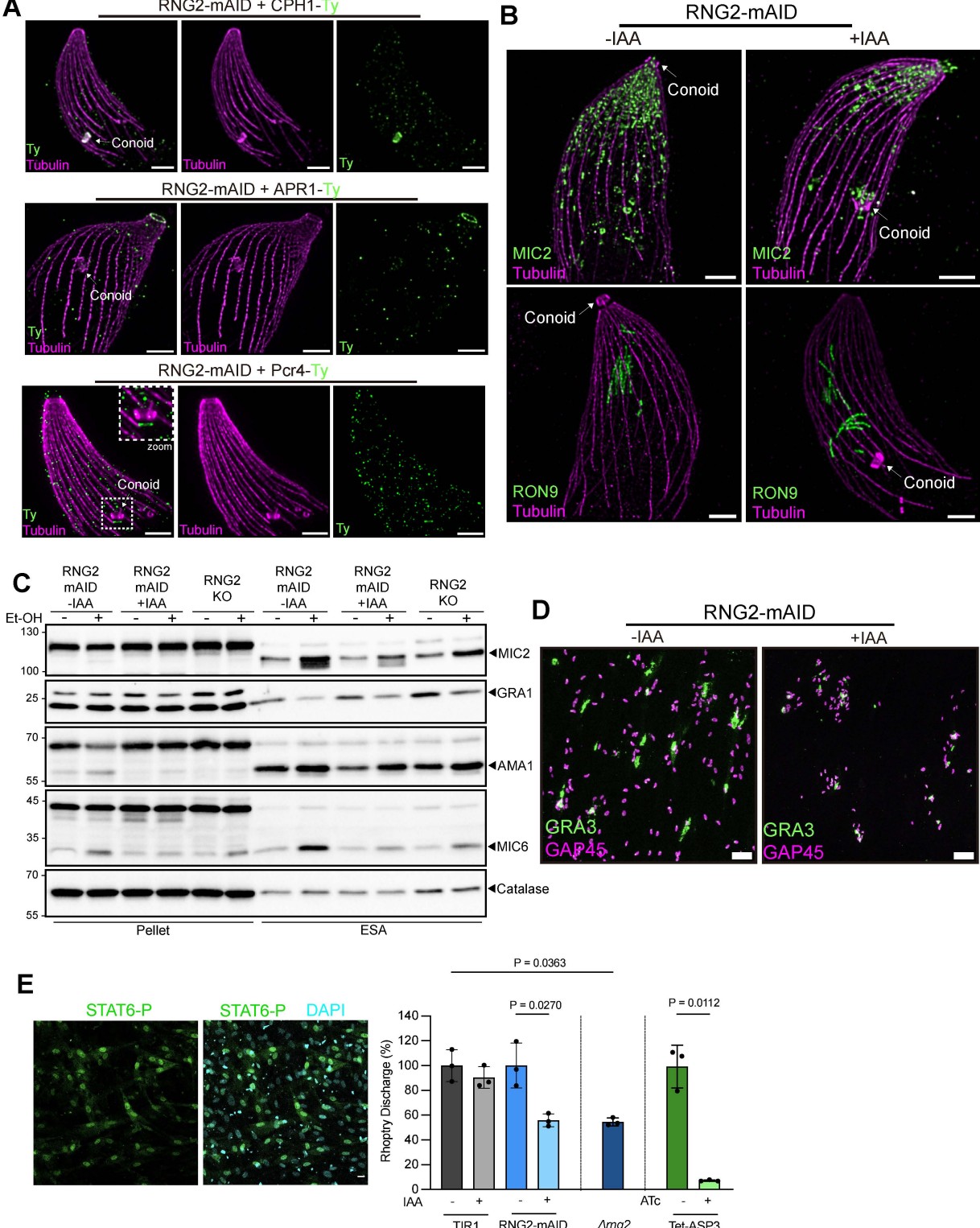

**Fig 5. Phenotypic consequences of the conoid detachment. (A)** U-ExM images of CPH1 (conoid marker) and APR1 (APR marker) with the conoid detached from the APR. Scale bar = 3 μm. **(B)** U-ExM of secretory organelles, with MIC2 used as a microneme marker and RON9 used as a rhoptry neck marker. Images taken with the conoid attached or detached from the APR. Scale bar = 3 μm **(C)** Western blot analysis of ethanol-triggered microneme

secretion. ESA, Excreted secreted antigens. **(D)**. Representative images of BIPPO-induced egress. Scale bar = 20 μm. **(E)** Left panel: representative images of STAT6-P-positive cells with a Tir1 strain. Right panel: Quantification of rhoptry secretion was performed to assess STAT6 phosphorylation. At least 200 cell nuclei were counted for data plotting, $n = 3$. ATc = anhydrotetracycline. The data underlying this figure can be found in S2 Data.

detachment, including the PCRs, disrupted apical F-actin polymerization and mislocalized GAC, preventing glideosome assembly and causing a profound motility defect.

## Domain mapping of RNG2 interfaces with the conoid and APR

To investigate the regions of RNG2 involved in conoid tethering, the Δ*rng2* parasite line was complemented with truncated versions of RNG2 expressed as a second copy under the control of the tubulin promoter (Fig 6A). WB analysis confirmed that each variant was expressed at the expected size and revealed processed forms similar to those observed for the endogenous protein (Fig 6B). IFA and U-ExM analyses of all truncated proteins confirmed their localization to the APR, with an additional cytoplasmic signal likely resulting from overexpression (Figs 6C and S6A). Plaque assays assessing the fitness of the complemented lines showed that the WT, ΔNTer, and ΔTropo variants fully restored parasite fitness after deletion of endogenous RNG2 (Figs 6D and S6A). In contrast, the ΔCTer and ΔN+CTer variants only partially restored parasite fitness, indicating that the C-terminal region of RNG2 is critical for its physiological function, whereas the N-terminal region and the central tropomyosin-like domain appear dispensable (Figs 6D and S6A). Although the ΔCTer- and ΔN+CTer- complemented strains displayed a mild invasion defect, the WT and ΔTropo strains fully rescued the invasion phenotype. The ΔNTer variant, however, showed a slight but significant reduction in invasion (~10%) (Fig 6E).

To assess the ability of each construct to correctly maintain the conoid at the apical pole, conoid detachment was assessed by U-ExM after activation with BIPPO. The wild-type and ΔTropo variants successfully complemented the deple-tion of endogenous RNG2, resulting in very few detached conoids. In contrast, 50% of conoid detachment was scored in ΔCTer and ΔN+CTer variant-complemented strains (Fig 6F). Although the ΔNTer strain did not exhibit a significant fitness defect, approximately 20% of parasites displayed a detached conoid, correlating with the mildly impaired invasion pheno-type (Fig 5F). Following conoid extrusion, the protein variants either remained tethered to the APR, as observed with the ΔNTer variant, or were associated with the floating conoid, as seen in the ΔCTer and ΔN+CTer strains. These observations confirm that the N-terminal region of RNG2 associates with the conoid, while the C-terminal region mediates interaction with the APR (Fig 6G).

Two additional RNG2 variants were engineered to investigate the internal coiled-coil region: one with an extended central region containing six tropomyosin domain repeats (RNG2-6Tropo), and another shorter variant in which most of the internal sequence was deleted (RNG2-ΔCenter) (Fig 6H). Both variants were dually tagged with Myc-tags and Ty-tags at their N- and C-terminal ends, respectively. WB analysis confirmed that both variants were expressed correctly and exhib-ited several processed forms, particularly in the extended RNG2 protein (Fig 6I). Comparison of the blots indicated the presence of several fragments of identical size between the two variants, suggesting that a major N-terminal fragment of approximately 110 kDa and two key C-terminal fragments of 60 and 40 kDa were produced (Fig 6I). Both variants accu-mulated at the apical pole, as shown by IFA (S6B Fig). While the 6Tropo variant fully complemented the Δ*rng2* strain, the ΔCenter strain failed to do so, leading to conoid detachment (Figs 6J–6L and S6B). The 6Tropo variant dually localized to the conoid and APR (Fig 6M). In contrast, RNG2-ΔCenter localized to both the conoid and APR when detected with antibodies against either the N- or C-terminal regions, suggesting that the shortened protein is too short to span the gap between the two structures (Fig 6M). These results indicate that while the N- and C-terminal extremities of RNG2 are suffi-cient for binding the conoid and APR, respectively, the central region is essential for tethering the conoid at the apical pole.

Noting that three main truncation products of RNG2 during purification from insect cells, we asked whether these iso-forms might be physiologically relevant. The three truncated variants (155, 90, and 60 kD) were expressed in Δ*rng2* strain

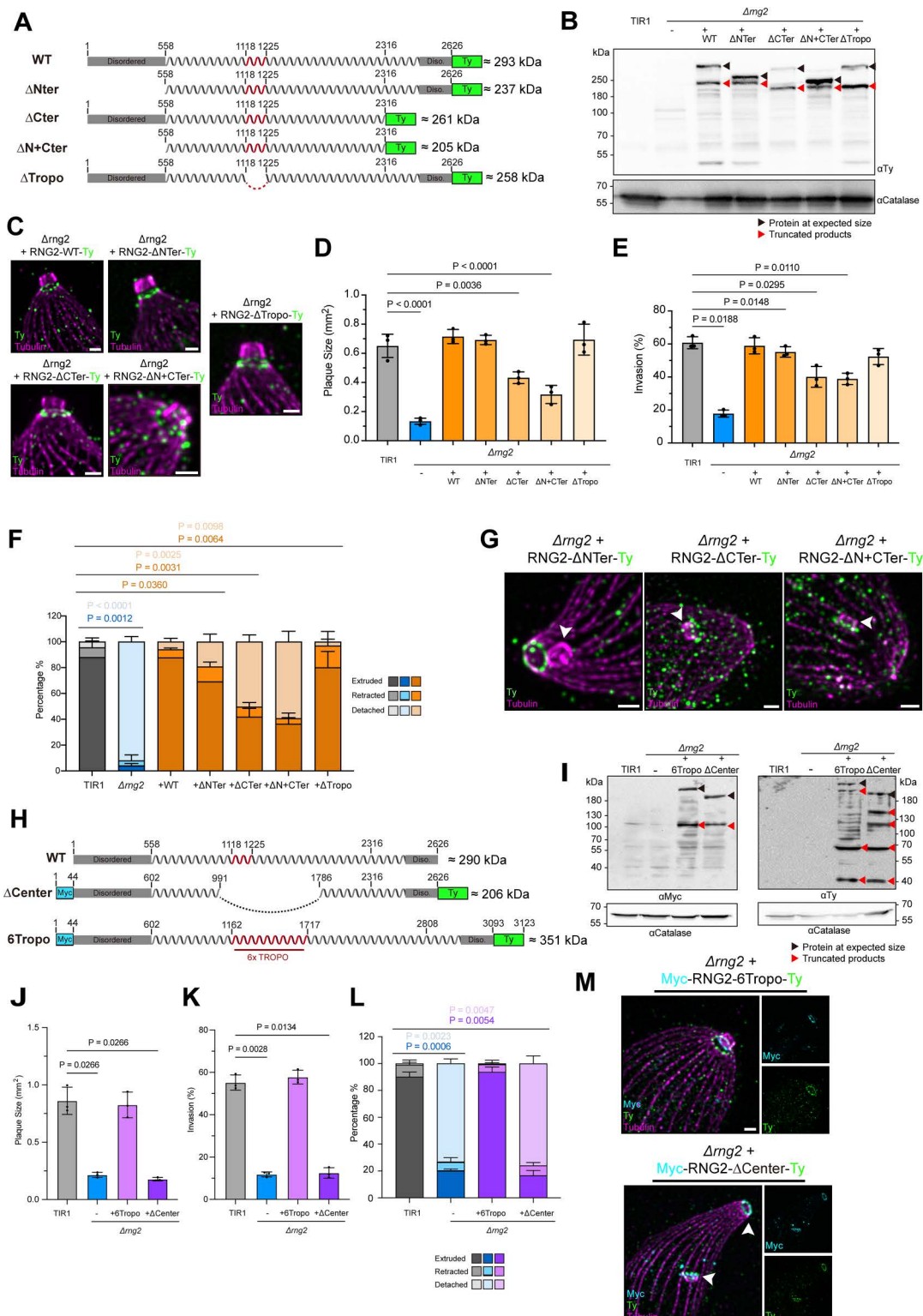

**Fig 6. Investigation of RNG2 subdomain functions. (A)** Representation of the RNG2 variants expressed in the Δ*rng2* background. **(B)** Western blot analysis of the expression of RNG2 variants expressed in the Δ*rng2* background. **(C)** U-ExM images of the localization of the RNG2 variants. Scale bar = 1 μm. **(D)** Quantification of the plaque sizes created by the different RNG2 strains via a plaque assay. Ten plaques were measured per biological

replicate. Three biological replicates were evaluated. The data underlying this figure can be found in S2 Data. **(E)** Quantification of the percentage of invasion by the different RNG2 strains in the invasion assay. One hundred parasites were measured per biological replicate. Three biological replicates were evaluated. The data underlying this figure can be found in S2 Data. **(F)** Quantification of the percentages of conoid retraction, extrusion, and detachment by the different RNG2 strains in the conoid detachment assay. One hundred parasites were measured per biological replicate. Three biological replicates were evaluated. The data underlying this figure can be found in S2 Data. **(G)** U-ExM images of the localization of the RNG2 variants under conditions in which the conoid detached from the apical pole. Scale bar = 1 µm. **(H)** Representation of the second round of RNG2 variants expressed in the Δ*rng2* background. **(I)** Western blot analysis of the expression of the second round of RNG2 variants expressed in the Δ*rng2* background. **(J)** Quantification of the plaque sizes created by second-round RNG2 variants via a plaque assay. Ten plaques were measured per biological replicate. Three biological replicates were evaluated. The data underlying this figure can be found in S2 Data. **(K)** Quantification of the percentage of invading cells in response to the second round of RNG2 overexpression via an invasion assay. One hundred parasites were measured per biological replicate. Three biological replicates were evaluated. The data underlying this figure can be found in S2 Data. **(F)** Quantification of the percentage of conoid retraction/extruded/detaching by the second round of the RNG2 variant in the conoid detachment assay. One hundred parasites were measured per biological replicate. Three biological replicates were evaluated. **(L)** U-ExM images of the localization of the second round of RNG2 variants under conditions in which the conoid detached from the apical pole. Scale bar = 1 µm. The data underlying this figure can be found in S2 Data.

to assess their function (Fig 7A). All three variants bear the epitope tags both at N-terminal (Myc) and C-terminal (Ty), and WB analysis confirmed the expression of three variants (Fig 7B), located at apical region of the parasites. While the 155 kD variant exhibited dotted accumulated signal in the cytosol similar to the wild-type, expression of 90 and 60 kD smeared within the cytosol (S6C Fig). U-ExM revealed that all three truncated variants localized to the APR, with their N- and C-termini colocalized, unlike the dual localization observed for endogenous RNG2 (Fig 7C). Functional assessment by plaque assay (Figs 7D and S6C), invasion assay (Fig 7E), and conoid anchoring (Fig 7F) showed that none of the three isoforms could complement the loss of endogenous RNG2.

## RNG2 needs to be intact to tether the conoid at the apical pole

Given the extensive proteolytic processing of RNG2, both in a heterologous system and in the parasite, and its tendency to oligomerize, we hypothesized that the processed products might form oligomers in the parasite and provide flexible length to accommodate rapid distance changes during conoid extrusion and retraction. To test whether different fragments of RNG2 can interact and thereby restore function, RNG2-mAID-HA parasites stably expressing the RNG2-ΔCenter variant were transiently transfected with the 155 kDa fragment that spans the deleted region (Fig 7G). After 48 h of IAA treatment to deplete endogenous RNG2-mAID-HA, conoid extrusion was induced using BIPPO, and the proportion of conoids correctly positioned at the apical pole was assessed. As previously shown, ~80% of conoids were detached in the RNG2-mAID-HA strain, while only ~20% were detached when a full-length copy of RNG2 was transiently expressed (Fig 7H). As observed earlier (Fig 7F), transient expression of the 155 kDa fragment alone did not rescue conoid anchoring, with ~80% of conoids remaining detached. In contrast, transient transfection of full-length RNG2 into the RNG2-mAID-HA strain stably expressing RNG2-ΔCenter drastically reduced the number of detached conoids. These results indicate that the fragments do not functionally cooperate or physically interact in a way that restores RNG2 function, similar to what was observed in vitro.

As a second approach to assess whether RNG2 proteolytic processing is functionally important, we designed a construct in which a P2A "skip" peptide [34,35] was inserted within the tropomyosin domain of RNG2. This insertion causes ribosomal skipping during translation, leading to expression of two separate RNG2 fragments (RNG2-P2A WT). In parallel, we generated a control construct in which key residues of the P2A sequence were mutated to prevent skipping [34], thereby preserving RNG2 as a single polypeptide (RNG2-P2A MUT). Both constructs were stably expressed in the RNG2-mAID-HA line to assess whether the co-expressed fragments could functionally complement RNG2 depletion (Fig 7I). WB analysis confirmed co-expression of both N- and C-terminal fragments from the RNG2-P2A WT construct, while the RNG2-P2A-MUT remained unsplit and displayed the same proteolytic processing as the endogenous RNG2 (Fig 7J). Upon BIPPO-induced conoid extrusion, the strain expressing RNG2 P2A-WT exhibited detachment of the conoid

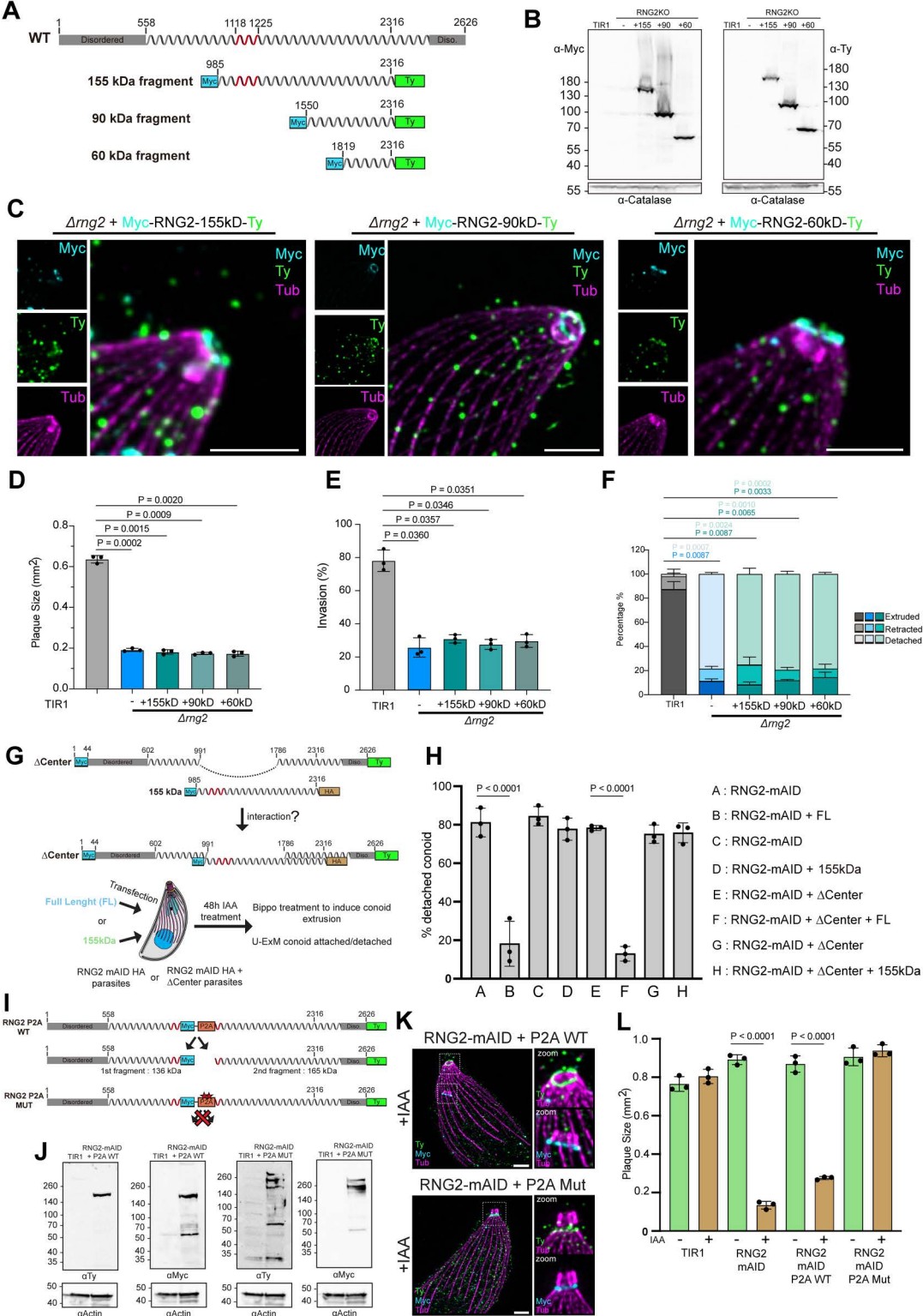

**Fig 7. RNG2 is functional only as intact, full-length protein. (A)** Representation of the RNG2 three processed products originally identified in insect cells (155kDa, 90kDa, 60kDa) expressed in the Δ*rng2* background. **(B)** Western blot analysis of the expression of RNG2 variants expressed in the Δ*rng2*

background. **(C)** U-ExM images of the localization of the RNG2 variants. Scale bar = 3 µm. **(D)** Quantification of the plaque sizes created by the different RNG2 strains via a plaque assay. Ten plaques were measured per biological replicate. Three biological replicates were evaluated. The data underlying this figure can be found in S2 Data. **(E)** Quantification of the percentage of invasion by the different RNG2 strains in the invasion assay. One hundred parasites were measured per biological replicate. Three biological replicates were evaluated. The data underlying this figure can be found in S2 Data. **(F)** Quantification of the percentages of conoid retraction, extrusion, and detachment by the different RNG2 strains in the conoid detachment assay. One hundred parasites were measured per biological replicate. Three biological replicates were evaluated. The data underlying this figure can be found in S2 Data. **(G)** Design of the experiment to test if different fragments of RNG2 can cooperate to form a functional tether. **(H)** Conoid detachment assessed by U-ExM in transiently transfected parasites treated for 48h under auxin to down-regulate the endogenous RNG2-mAID. The data underlying this figure can be found in S1 Data. **(I)** Representation of the skip peptide (P2A) variants expressed in the RNG2-mAID-HA strain. **(J)** Western blot analysis of the expression and skipping pattern verifying the successful skipping of ribosome on the P2A sequence. **(K)** U-ExM images of auxin treated parasite to deplete the endogenous RNG2, showing the localization of both P2A variants and their ability to maintain the conoid at the apical pole. **(L)** Quantification of the plaque sizes created by RNG2 P2A variants via a plaque assay. Ten plaques were measured per biological replicate. Three biological replicates were evaluated. The data underlying this figure can be found in S2 Data.

from the apical pole. The N-terminal fragment of RNG2 localized to the displaced conoid, while the C-terminal fragment remained associated with the APR (Fig 7K). In contrast, in the strain expressing the RNG2 P2A-MUT, the protein localized between the conoid and APR and rescued the phenotype of conoid detachment (Fig 7K). In line with this observation, the RNG2-P2A-WT failed to complement by plaque assay, while the RNG2-P2A-MUT fully complemented the depletion of the endogenous protein (Fig 7L). Collectively, these results demonstrate that full-length RNG2 is required to tether the conoid at the apical pole.

## Discussion

The conoid serves as a critical hub for assembling all the molecular machinery necessary for parasite invasion. This dynamic cytoskeletal element orchestrates the discharge of secretory organelles and the assembly of the actomyosin-based glideosome, which powers parasite motility. Originally believed to be exclusive to cyst-forming parasites, the conoid has recently been found in a broader range of apicomplexans, including *Plasmodium* and *Cryptosporidium* species [17,36–38].

In *T. gondii*, the conoid undergoes calcium-dependent extrusion and retraction, powered by MyoH [11] and passing through the APR. This process is crucial for controlling parasite motility by directing actin filaments into the pellicular space, leading to MyoA-driven forward motion [11]. The integrity of the APR is also critical to control the F-actin flux [22]. Despite this understanding, the factors responsible for maintaining the conoid at the apical pole and facilitating its movement through the APR remain unclear. The observation of BFs acting as tethers between the APR and conoid identified the first physical link between these structures; however, the molecular entities associated to this structure are unknown [24].

RNG2, a large coiled-coil protein conserved in Coccidia with orthologs present in *Neospora caninum*, *Sarcocystis neurona*, and *Eimeria acervulina* (NCLIV_019030; SN3_00201395; EAH_00027490), has emerged as a strong candidate for APR attachment due to its dual localization at the APR and conoid [23]. In vitro studies revealed that RNG2 is prone to proteolytic cleavages and oligomerization. Similar processing events are also observed in the parasite. High-resolution imaging based on U-ExM and iU-ExM and IEM detected RNG2 as 22 puncta at and between the APR and conoid, supporting its role as part of the BFs. Of relevance, RNG2 depletion caused conoid detachment upon BIPPO stimulation, emphasizing its tethering role. Notably, the detachment occurs following extrusion and is blocked by CD, indicating that RNG2 is crucial for maintaining conoid positioning during and after extrusion.

Although microneme secretion was partially affected in RNG2-depleted parasites, conoid extrusion was not affected. This is consistent with previous observations reporting that the two events are not connected, with MyoH-depleted parasites having conoid extrusion defects but leaving microneme secretion unaffected [11]. Remarkably, some micronemes clearly remain associated to the detached conoid (Fig 5B), suggesting intrinsic microneme-binding properties within the conoid. In this context, three major regulators of microneme apical positioning and exocytosis, namely, HOOK, FTS, and

HIP, were previously shown to accumulate at the apical pole [39]. Although their precise localization within the apical pole remains unclear, investigating RNG2-depleted parasites could shed light on whether these factors reach the conoid, potentially explaining how micronemes remain associated with the detached conoid.

RNG2 depletion did not completely compromise parasite fitness, resulting in only a 50% reduction in invasion. Although microneme secretion was only mildly affected, RNG2-depleted parasites displayed motility defects due to conoid detachment, which impaired the translocation of F-actin required for motility.

Proper positioning of the rhoptries is essential for their discharge and was disrupted in RNG2-depleted parasites, explaining the significant reduction in rhoptry secretion observed in the absence of RNG2. However, the phenotype of RNG2-depleted parasites is not as severe as that of ARO-deficient parasites in which rhoptries are mispositioned during their biogenesis [40,41]. This suggests that proper rhoptry positioning in intracellular parasites depleted in RNG2 preserves the potential for invasion, provided that activated parasites are in immediate contact with a host cell. Recent studies have demonstrated that during rhoptry discharge, two rhoptries dock at the AV, located at the parasite plasma membrane [42]. Given the proximity of the AV to the PCRs, it was important to investigate AV positioning following conoid detachment. The AV stays correctly positioned apically in RNG2-depleted parasites, highlighting its strong interaction with the parasite plasma membrane mediated by the rhoptry secretion apparatus [13,29,42].

A comprehensive mutagenesis of RNG2 led to the mapping of the N-terminal region important for binding to the conoid and the C-terminal region anchoring RNG2 to the APR. Whether RNG2 associates directly with tubulin fibers or through intermediary proteins linked to the structure is currently unknown. The conoid detachment observed upon APR1 depletion suggests a possible interaction with RNG2 [19].

Importantly, the biochemical analysis of recombinant RNG2 revealed that the protein is prone to proteolytic cleavage and, owing to its large coiled-coil domains, readily forms oligomers. The observed homo-oligomerization of processed fragments under co-expression conditions is consistent with the known behavior of large coiled-coil domains, which

preferentially form homo-oligomers of identical length. Such assemblies are thermodynamically favored over hetero-oligomers composed of fragments of differing lengths [25]. This is a noteworthy observation, as the propensity for self-association likely reflects RNG2's behavior in the parasite, where cleavage fragments may also preferentially assemble into homo-oligomers. These oligomeric assemblies likely contribute to the fiber-like structures observed in vivo.

A series of RNG2 truncation mutants were tested in functional complementation assays to determine whether protein length is critical and whether processed fragments could assemble into concatemers of sufficient size. The results indicate that RNG2 must remain intact: while its size can be extended, truncation of RNG2 (ΔCenter) disrupts function. Expression of individual partially overlapping fragments in trans failed to complement shorter versions of the protein, suggesting that proper function requires a continuous, full-length molecule capable of bridging the conoid and APR. The failure of trans-complementation can be explained by the thermodynamic disfavoring of hetero-oligomeric interactions between fragments of different lengths, implying that only full-length RNG2 serves as a functional tether. We can, however, hypothesize that association of truncation fragments at the APR may contribute to filament rigidity, stabilizing the conoid in its extruded position, allowing protein secretion and parasite motility.

A second approach was used to support the proteolytic processing of RNG2 is not sufficient to maintain its function in conoid tethering. While expression of split RNG2 fragments via a P2A "skip" peptide allowed co-expression of the N- and C-terminal portions, these fragments failed to anchor the conoid at the apical pole, leading to conoid detachment and loss of functional complementation. Even when the P2A-WT was co-transfected with the 155 kDa expressing plasmid, no complementation was observed. In contrast, preserving RNG2 as a full-length polypeptide with the P2A mutant, restored proper localization between the conoid and APR and fully rescued the phenotype. Through a combination of biochemical assays, genetic manipulations, and cell imaging, we propose a model in which RNG2 maintains the conoid at the apical pole through

strong interactions mediated by its N- and C-terminal domains, binding the conoid and APR, respectively (Fig 8). The pivotal role of full-length RNG2 in connecting the conoid to the APR enables dynamic transitions between extrusion and retraction that critically control motility and invasion.

## Materials and methods

### Parasite maintenance

*T. gondii* tachyzoites were amplified from HFFs (ATCC) in Dulbecco's modified Eagle's medium (DMEM, Gibco) supplemented with 5% fetal calf serum (FCS, Gibco), 2 mM glutamine, and 25 µg/ml gentamicin (Gibco). Parasites and HFFs were maintained at 37 °C with 5% $CO_2$.

### Generation of transgenic cell lines

**a. Generation of RNG2-mAID (triple epitope-tagged strain).** The gene map of RNG2 was retrieved from the ToxoDB website [43] (www.toxodb.org) via the accession number TGGT1_244470. A specific gRNA (guideRNA) was designed to target the 3′UTR of the endogenous locus via the EuPaGDT tool (www.grna.ctegd.uga.edu) [44]. The sequence of the gRNA (AAGAGAAATTGCCTTCATGT) was inserted into pU6-Universal (pU6-Universal was a gift from Sebastian Lourido, Addgene plasmid #52694) [45]. A repair fragment containing the mAID-HA cassette, and flanking regions was amplified via PCR via KOD polymerase (Novagen, Merck) via the oligos and template listed in S1 Data. Freshly egressed RHΔKu80ΔTir1 (called Tir1) tachyzoites were transfected via electroporation [46] with 40 µg pU6-Universal bearing the gRNA alongside 100 µL (2 PCRs of 50 µL) of repair fragment containing the mAID-HA cassette and homology regions for the gene. For enrichment of the transfected population, parasites carrying an HXGPRT cassette were selected with 25 mg/ml mycophenolic acid and 50 mg/ml xanthine. The parasites were then cloned and inserted into p96w, and the clonality of the population was assessed via integration PCR, as shown in S2A Fig.

To insert a 3Myc tag on the N-terminal side of the RNG2 gene, two gRNAs were designed to target the 5′UTR of the gene (CAACCGTTCCATGTAAGGCC) and a few base pairs after the START codon (TTCGGGAGACGTTTCTCCTA). The two gRNAs were inserted into the NdeI linearized plasmid listed in S1 Data via Gibson assembly. A repair fragment of approximately 500 bases containing the 3-Myc-Tag inserted just after the start codon and containing homology regions on each side was synthesized (g-block IDT DNA company). This repair fragment was inserted into the pCR-Blunt-II-Topo vector (Thermo Fisher, K270020) and then amplified via PCR via KOD polymerase in the same way as described previously.

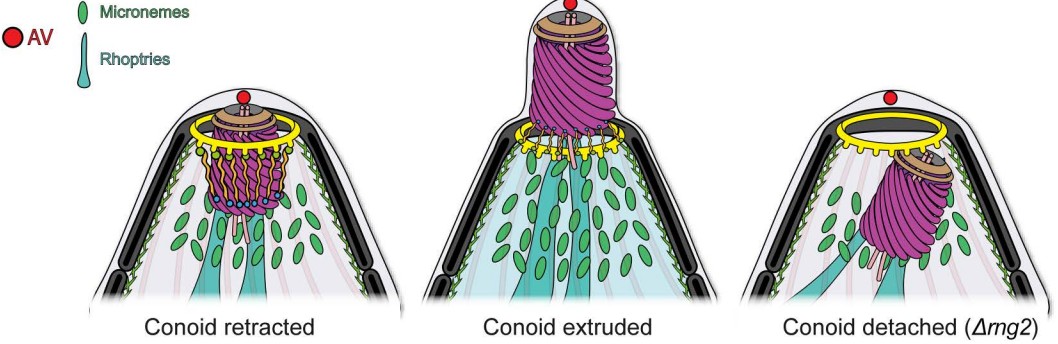

**Fig 8. Model RNG2 as a molecular tether for conoid apical anchorage.** Illustration of the consequences of RNG2 depletion on the apical complex structure of *Toxoplasma gondii* tachyzoites.

Freshly egressed clonal parasites bearing the mAID-HA cassette were transfected via electroporation as described previously and sorted via flow cytometry (Cas9-GFP). The selected parasites were then cloned and inserted into p96w, and the clonality of the population was assessed via PCR, as shown in S2A Fig.

To insert a 3Ty-Tag in the internal sequence of the RNG2 gene, a gRNA was designed to target this specific region (GAAGGGCAGAAGAACTT) and was inserted into the pU6-universal vector as described above. A repair fragment of approximately 500 bases containing the 3-Ty-Tag inserted just after the in-frame of the sequence and containing homology regions on each side was synthesized (g-block IDT DNA company). This repair fragment was inserted into the pCR-Blunt-II-Topo vector (Thermo Fisher, K270020) and then amplified via PCR via KOD polymerase in the same way as described previously. Freshly egressed clonal parasites bearing the N-Ter Myc-tag and mAID-HA cassette were transfected via electroporation as described previously and sorted via flow cytometry (Cas9-GFP). The selected parasites were then cloned and inserted into p96w, and the clonality of the population was assessed via PCR, as shown in

For all the assays involving AID-based conditional knockdown systems, protein depletion was achieved by adding 500 µM auxin (IAA) [47].

**b. Generation of the Δ*rng2* strain.** To generate the RNG2 knockout line, two gRNAs were designed to target the 5′UTR of the gene (CAACCGTTCCATGTAAGGCC) and the 3′UTR of the gene (TGTGTGTTCATACTCCCGTG). The two gRNAs were inserted into the NdeI linearized plasmid listed in S1 Data via Gibson assembly. A repair fragment containing the DHFR cassette as well as flanking regions on each side was amplified via PCR via KOD polymerase. Freshly egressed RHΔKu80ΔTir1 (called Tir1) parasites were transfected via electroporation as described previously. For enrichment of the transfected population, parasites carrying the DHFR cassette were selected with 1 µg/ml pyrimethamine.

The selected parasites were then cloned and inserted into p96w, and the clonality of the population was assessed via PCR.

**c. RNG2 variant expression at the UPRT locus.** The variant wild-type RNG2 gene was synthesized into the pFastBac vector between the BamHI and HindIII cloning sites (Genscript Company). The RNG2 gene was then subcloned and inserted into the pTub-UPRT-G13-Ty vector between the EcoRI and AvrII restriction sites and under the control of the tubulin promoter to generate the vector pTub-UPRT-RNG2-WT-Ty [48]. For each of the RNG2 variants, specific restriction sites were placed alongside the RNG2 gene sequence, allowing the insertion/deletion of the desired sequences.

For RNG2-ΔNter, pTub-UPRT-RNG2-WT-Ty was digested by the restriction enzyme EcoRI+SpeI and ligated with the primer pairs 10,817–10,818 to generate pUPRT-RNG2-ΔNter-Ty.

For RNG2-ΔCter, pFastBac-RNG2-WT was digested by the restriction enzyme NotI to generate pFastBac-RNG2-ΔCter. pFastBac-RNG2-ΔCter was then digested with EcoRI+XbaI and cloned and inserted into pTub-UPRT-G13-Ty, which was subsequently digested with EcoRI+AvrII to generate pTub-UPRT-RNG2-ΔCter-Ty. For RNG2-ΔN+Cter, pTub-UPRT-RNG2-ΔCter-Ty was digested by the restriction enzyme EcoRI+SpeI and ligated with primer pairs 10817/10818 to generate pTub-UPRT-RNG2-ΔN+Cter-Ty. For RNG2-ΔTropo, pFastBac-RNG2-WT was digested by the restriction enzymes SalI+XhoI, followed by ligation. The sequences were then digested by EcoRI+XbaI to clone into the EcoRI+AvrII digested vector pTub-UPRT-G13-Ty to generate pTub-UPRT-RNG2-ΔTropo-Ty.

For RNG2-ΔCenter, the RNG2-ΔCenter variant was synthesized and cloned, and inserted into the pTWIST vector. Cloning was performed via the digestion of pTWIST-RNG2-ΔCenter by EcoRI+HindIII and cloning into the vector pTub-MyoL-3Ty, which was digested by EcoRI+HindIII to generate pTub-UPRT-RNG2-ΔCenter.

For RNG2-6Tropo, 6xTropo fragments were synthesized and cloned, and inserted into the pTWIST-vector. The 6xTropo fragments were recycled by using the enzymes SalI+XhoI and cloned and inserted into the vector pFastBac-RNG2-WT, digested with SalI+XhoI to generate pFastBac-RNG2-6Tropo. pFastBac-RNG2-6Tropo was then digested by EcoRI+XbaI and cloned and inserted into pTub-UPRT-RNG2-ΔCenter digested by MfeI+NheI to generate pTub-UPRT-RNG2-6Tropo

For RNG2-155/90/60 fragment, corresponding fragment was amplified using primer pairs: 155 kDa (11347/11351), 90 kDa (11349/11351), 60 kDa (11350/11351). The template for PCR used was the pTUB-UPRT-RNG2-WT. The PCR

products were linearized with EcoRI+EcoRV (155 kD) or EcoRV+XmaI (90 kD/60 kD) and cloned into pTub-UPRT-G13-Ty vector linearized with same enzymes to make pTub-UPRT-Myc-155/90/60-Ty vector (3 different vectors).

For the skip peptide variants (P2A-WT and P2A Mut), the central region of the Myc-P2A-WT and Myc-P2A-MUT were synthesized into two pTWIST vectors between SpeI and XcmI cloning sites (Twist company). pTwist-RNG2-Myc-P2A-WT plasmid was digested by SalI and XhoI and the ligated with the pFastBac-RNG2-WT digested by the same enzymes to obtain the pFastBac-RNG2-Myc-P2A-WT. This plasmid was then digested by EcoRI and XbaI to transfer the RNG2-Myc-P2A-WT sequence into the pUPRT-G13-Ty digested by EcoRI and AvrII. This led to the obtention of the pUPRT-pTub-RNG2-Myc-P2A-WT-Ty plasmid that was used for transfection into the UPRT locus.

For the P2A-Mut, the pTwist-RNG2-Myc-P2A-Mut was digested by HindIII and NheI and ligated into the pUPRT-pTub-RNG2-Myc-P2A-WT-Ty digested by the same enzymes. This led to the obtention of the pUPRT-pTub-RNG2-Myc-P2A-Mut-Ty that was used for transfection into the UPRT locus.

**d. Construction of variants for the transient complementation.** For the transient complementation, the RNG2-155kDa-HA fragment, the Myc-RNG2-155kD-Ty cassette has been recycled through EcoRI+EcoRV from pTub-UPRT-Myc-155kDa-Ty vector as stated above and cloned into pTub-HX plasmid digested by same enzymes to generate pTub-HX-Myc-RNG2-155kDa-Ty vector. The pTub-HX-Myc-RNG2-155kDa-Ty vector was digested by XmaI+EcoRV to allow the integration of annealed oligo pairs 12253/12254. The annealed short fragment contains 2xHA tag and a stop codon, allowing the production of plasmid pTub-HX-Myc-RNG2-155kDa-HA. For the transient complementation of the full-length RNG2, the same process was performed to generate the pUPRT-pTub-RNG2-HA from the pUPRT-pTub-RNG2-TY.

For transfection, parasites (RNG2-mAID-HA or RNG2-mAID-HA+ΔCenter) were electroporated with 100 µg of plasmid encoding the selected RNG2 variant (FL or 155kDa) and treated with auxin during 48h. U-ExM procedure was then followed on extracellular parasites treated by BIPPO (15 min RT) to assess the state of the conoid.

**e. RNG2 variant cloning for insect cells expression.** The variant wild-type RNG2 gene was synthesized into the pFastBac vector between the BamHI and HindIII cloning sites (Genscript Company). Specific restrictions sites have been placed within the RNG2 sequence to allow mutagenesis of the protein. To generate the pFastBac-RNG2-ΔNter, the pFastBac-RNG2-WT was digested by NheI and SpeI and ligated on itself, allowing the excision of the N-terminal region. To generate the pFastBac-RNG2-Cter, the pFastBac-RNG2-WT was digested by Not1 and ligated on itself, allowing the excision of the C-terminal region. The same procedure was performed with the pFastBac-RNG2-ΔNter to generate the pFastBac-RNG2-ΔN+Cter. Finally, to generate the pFastBac-RNG2-ΔTropo, the pFastBac-RNG2-WT was digested by SalI and XhoI and ligated on itself, allowing the excision of the tropomyosin domain.

The RNG2-155/90/60 fragments correspond to the following amino acid boundaries, respectively: 155 kDa (985–2290), 90 kDa (1524–2290), 60 kDa (1739–2290). For RNG2-155/90/60 recombinant protein expression in insect cells, the corresponding fragments were amplified using primer pairs: 155 kDa (11352/11348), 90 kDa (11353/11355), 60 kDa (11354/11355). The template for PCR used was the pFastBac-His-RNG2-ΔCter-Strep. The PCR products were digested with NheI+XhoI (155 kDa) or with EcoRI+NotI (90, 60 kDa) and cloned into the digested pFastBac-His-RNG2-ΔCter-Strep vector, digested with the same enzymes to make the pFastBac-His-155/90/60-Strep vectors (3 different vectors). All fragments possess both a N-terminal His-tag and a C-terminal Strep tag.

To test the potential interaction between the different fragments, a RNG2 90 kDa fragment of 90 kDa was synthesized that lacks the N-terminal His-tag (RNG2-Strep 90 kDa). Cloning of recombinant RNG2-Strep 90 kDa was done by amplifying template pFastBac-His-RNG2-Strep 90 kDa using primer pair 11547/11548. The PCR product and template were digested with BamHI+AatII and ligated to obtain the final RNG2-Strep 90 kDa vector.

## Baculovirus-infected Insect cell expression and purification

All recombinant RNG2 proteins were expressed in Sf9 insect cells and purified following a similar protocol. All the plasmids encoding RNG2 possessed an N-terminal His10-tag followed by a TEV recognition sequence and an HRV-3C

recognition sequence, followed by a TwinSTREP tag. Baculoviruses were generated following standard procedures, and the cells were infected for 48 h at 27 °C in a shaking incubator. Purifications were performed via a combination of STREP-tactin affinity purification and size-exclusion chromatography. Briefly, a pellet from 600 ml of cells was resuspended in 120 ml of lysis buffer (PBS supplemented with 800 mM NaCl, 1 mM EDTA, 0.1% Triton X-100, and 3 mM beta-mercaptoethanol) and lysed by shear force using a LM-20 microfluidizer set at 20,000 psi. The lysate was centrifuged at 35,000$g$ for 35 min at 4 °C, and the supernatant was applied to a 5 ml STREP-TACTIN XT 4Flow column (IBA) at 0.5 ml/min. The column was washed with 50 ml of lysis buffer and eluted with 15 ml of lysis buffer supplemented with 50 mM biotin. The eluted protein was concentrated to 1 ml via a 30 MWCO Amicon concentrator and loaded on a Superdex 200 10/300 GL column at 22 °C previously equilibrated in PBS supplemented with 3 mM DTT. Fractions were analyzed via SDS–PAGE, pooled, and flash frozen in liquid nitrogen before being stored at −80 °C.

### Expansion microcopy (U-ExM)

For this study, the expansion microscopy protocol applied to *T. gondii* tachyzoites was followed as previously described [28]. Briefly, parasites resuspended in warm PBS containing 10 μM BIPPO were seeded on 12 mm coverslips precoated with poly-D-lysine (Gibco) for 10 min. After excess PBS was removed, the coverslips were incubated in PBS containing 0.7% formaldehyde and 1% acrylamide for 3 h at 37 °C. Polymerization of the expansion gel was performed on ice and contained a monomer mixture (19% sodium acrylate/10% acrylamide/0.1% bis-acrylamide), 0.5% ammonium persulfate, and 0.5% tetramethylenediamine. Fully polymerized gels were denatured at 95 °C for 90 min in denaturation buffer (200 mM SDS, 200 mM NaCl, 50 mM Tris, pH = 9) and expanded in pure $H_2O$ overnight.

The next day, the expansion ratio of the fully expanded gels was determined by measuring the diameter of the gels. The well-expanded gels were shrunk in PBS and stained with primary and secondary antibodies diluted in freshly prepared 2% PBS/BSA at 37 °C for 2 h. Three washes with PBS/0.1% Tween for 10 min were performed after primary and secondary antibody staining. The stained gels were expanded again in pure $H_2O$ overnight for further imaging. All U-ExM images used in this study were acquired via Leica TCS SP8 microscope with the lens HC PL Apo 100×/1.40 Oil CS2. Images were taken with Z-stacks and deconvolved with the built-in setting of Leica LAS X. Final images were processed with ImageJ, and the maximum projected images are presented in this study.

### Iterative Ultrastructure Expansion Microscopy (iU-ExM)

Iterative ultrastructure expansion microscopy was applied as described [26]. Briefly, parasites resuspended in warm PBS containing 10 μM BIPPO were seeded on 12 mm coverslips precoated with poly-D-lysine (Gibco) for 10 min. After excess PBS was removed, the coverslips were incubated in PBS containing 0.7% formaldehyde and 1% acrylamide for 3 h at 37 °C. The first monomer solution was added to fill in the space of the gelation chamber. Following an incubation of 15 min on ice, an incubation at 37 °C for 45 min was completed. The formed gels were transferred into the denaturation buffer and incubated at 85 °C for 90 min. The gels were placed in water to expand 5–6 times their size and were shrunk with PBS to apply primary and secondary antibodies. The gels were bathed in water again and were cut into pieces that were incubated 3 times for 10 min on an agitating platform with activated neutral gel. The gels were dried by sliding it onto the microscope slide. A coverslip was placed on top, and the chamber was incubated at 37 °C for 1 h. The gel was bathed in the combination of acrylamide and formaldehyde for overnight at 37 °C under agitation. The following day, the gel was washed twice with 1× PBS for 30 min before applying the 3rd monomer solution to achieve an expansion factor of approximately 16×. The gel was washed 3 times for 10 min under agitation on ice and was dried on the microscope slide. A coverslip was placed on top within a chamber and incubated for 1 h at 37 °C. The gel was bathed in 200 mM NaOH solution for 1 h under agitation at RT and was washed in 1× PBS for 20 min

until the pH reaches 7. The last round of expansion was carried out with water, and cells were imaged using Leica TCS SP8 microscopy with the lens HC PL Apo 100×/1.40 Oil CS2. Images were taken with Z-stack and deconvolved with the built-in setting of Leica LAS X or with Huygens software. Final images were processed with ImageJ, and the maximum projected images were presented in this study.

## Electron microscopy

Extracellular tachyzoites were prepared in the same manner as previously described [28]. Briefly, the extracellular parasites were pelleted in PBS. Conoid extrusion was induced by incubation with 40 µl of BIPPO in PBS for 5 min at 37 °C. A 4 µl sample was applied to a glow-discharged 200-mesh Cu electron microscopy grid for 10 min. The excess sample was removed by blotting with filter paper and immediately washed three times in double-distilled water. Finally, the sample was negatively stained with a 0.5% aqueous solution of phosphotungstic acid for 20 s and air-dried. Electron micrographs of parasite apical poles were collected with a Tecnai 20 transmission electron microscope (FEI, the Netherlands) operated at an acceleration voltage of 80 kV and equipped with a side-mounted CCD camera (MegaView III, Olympus Imaging Systems) controlled by iTEM software (Olympus Imaging Systems).

## Immunoelectron microscopy

The isolation of the cytoskeleton from tachyzoites of *T. gondii* was performed as previously reported [24]. Briefly, a suspension of tachyzoites ($1 \times 10^6$/ml) suspended in PHEM solution (10 mM HEPES, 10 mM EGTA, 1 mM $MgCl_2$, 50 µg/ml N-Tosyl-L-phenylalanine chloromethyl ketone, 50 µg/ml Nα-p-Tosyl-L-lysine chloromethyl ketone, and 17.4 µg/ml phenylmethylsulfonyl fluoride, pH 6.9) was exposed to 0.1% Triton X-100 in PHEM for 1 min at RT. The cytoskeleton fraction was centrifuged at 200,000*g* for 15 min at 4 °C. Under such conditions, most of the cytoskeletons were integrated into the typical crescent shape of tachyzoites. To isolate the cytoskeletons in a cartwheel-like conformation, tachyzoites were exposed to Triton X-100 solution for 3 min. For IEM, cytoskeletons were deposited on Ni grids covered with a formvar film (Polysciences, Warrington, PA, USA), fixed with 1% PAF in PHEM, blocked with 0.1% BSA, and then incubated for 2 h with anti-RNG2 (1:100 dilution) diluted in PHEM inside a humid chamber at RT. After washing, the samples were incubated for 2 h with IgG goat anti-rabbit IgG antibodies (ZYMED Laboratories, San Francisco, CA) coupled to 10 nm colloidal gold particles diluted in PHEM, washed with PHEM, counterstained with uranyl acetate, and examined via TEM. As a negative control, the cytoskeleton was incubated with preimmune serum and then with a secondary antibody coupled to gold particles.

## Cryo-electron tomography

Freshly egressed parasites pretreated 48 h ±IAA were harvested and washed with PBS. Conoid extrusion was induced by incubating the parasites in warm 10 µM of BIPPO in PBS for 10 min. Formaldehyde 4% was then used to fix the samples. Fixation was quenched with PBS-Glycine, washed with PBS, and kept on ice. 3 µL of sample (~$4 \times 10^6$ tachyzoites) were loaded on EM grid for plunge freezing into a liquid ethane on an EM GP2 automatic plunger (Leica Microsystems, Wetzlar, Germany) after 4–5 sec front blotting. Images were collected on a Thermo Fisher 300 kV Titan Krios G4 Cryo-TEM paired with the Thermo Scientific Falcon 4 Direct Electron Detector. Tilt-series were collected with a dose-symmetric scheme ranging from −55° to +55° with 2° increments at a magnification of 53000X, a pixel size of 2.42 Å, and a defocus range of −6 to −8 µm. The cumulative dose of each tilt-series was 140 e⁻/Å². Tilt series were motion corrected and CTF estimated using Warp, and tomograms aligned and reconstructed with AreTomo software [49,50]. Segmentation was manually generated in IMOD software, and videos were created with UCSF ChimeraX [51,52]. Quantifications were done in IMOD. In total, 23 tomograms were analyzed for each condition.

## Plaque assay

HFF monolayers were infected with a serial dilution of *T. gondii* tachyzoites and grown for seven days at 37 °C. The cells were fixed with paraformaldehyde-glutaraldehyde for 10 min, followed by neutralization with 0.1 M PBS/glycine. The fixed monolayer was then stained with crystal violet for 2 h and then washed three times with PBS.

## Conoid extrusion assay

To induce conoid extrusion, freshly egressed tachyzoites were incubated with 10 μM BIPPO for 10 min at room temperature. The parasites were then transferred to poly-D-lysine-coated coverslips, followed by the ultrastructure expansion microscopy protocol as described previously. The conoid status of a minimum of 100 parasites was determined for quantification. The data presented in this study were acquired from three biological replicates.

## Invasion assay

Coverslips covered with the HFF monolayer were infected with *T. gondii* tachyzoites, centrifuged at 1,200$g$ for 1 min, and incubated at 37 °C for 30 min. Infected HFFs were fixed with paraformaldehyde-glutaraldehyde and neutralized with PBS/0.1 M glycine. The fixed samples were blocked with 5% PBS-BSA for 20 min at room temperature, followed by 1 h of staining with anti-SAG1 antibody and three washes in PBS. The stained samples were fixed again with 1% formaldehyde for 7 min and permeabilized with PBS/0.2% Triton X-100 for 20 min. The samples were then stained with anti-GAP45 antibody and secondary antibodies. At least 100 parasites per condition were counted to determine the invasion ratio. The data presented in this study were from three biological replicates.

## Microneme secretion assay

Freshly egressed parasites (syringed out in the case of parasites with severe egress defects) were washed twice in warm DMEM, pelleted, and resuspended in media containing 10 μM BIPPO. The suspensions were incubated at 37 °C for 15 min and centrifuged at 2,000$g$ to allow separation of the pellet/supernatant (ESA) fraction. The isolated ESA fractions were further centrifuged to remove any remaining cell debris. All the samples were subjected to western blot analysis with anti-MIC2, anti-MIC6, anti-AMA1, anti-Catalase, and anti-GRA1 antibodies.

## Egress assay

Fresh tachyzoites were used to infect HFF on coverslips, which were subsequently cultured for 30 h at 37 °C. Infected cultures were incubated with DMEM containing BIPPO (10 μM) for 10 min at 37 °C, followed by PFA/Glu fixation and neutralization with 0.1 M PBS/glycine. The coverslips were stained as described previously with the anti-GAP45 antibody and anti-GRA3. At least 100 vacuoles per condition were counted. The data presented in this study were acquired from three biological replicates.

## Gliding trail assay

Freshly egressed tachyzoites were resuspended in warm DMEM containing 10 μM BIPPO, seeded on 12 mm coverslips coated with poly-L-lysine, and centrifuged at 1,200$g$ for 2 min. The plates with coverslips were incubated for 30 min at 37 °C and fixed/stained as described. The anti-SAG1 antibody was resuspended in 2% PBS/BSA, and the corresponding fluorescent secondary antibody was used for visualization by imaging.

## STAT6-P rhoptry secretion assay

A total of $5 \times 10^6$ freshly egressed parasites were harvested and resuspended in 250 μl of DMEM for infection of one coverslip seeded with an HFF monolayer. After 30 s of centrifugation at 1,000$g$, the coverslips were incubated at 37 °C

for 20 min, followed by 8 min of fixation in ice-cold methanol (MeOH). After 30 min of blocking with 5% PBS/BSA, the coverslips were incubated overnight at 4 °C with a STAT6-P antibody. The next day, the coverslips were visualized with a fluorescent secondary antibody and DAPI. The rhoptry discharge efficiency was determined by the ratio of the number of STAT6-P-positive cell nuclei to the total number of nuclei (visualized by DAPI). At least 200 cell nuclei were counted for quantification. The data presented in this study were acquired from three biological replicates.

## Western blot analysis for monitoring auxin-induced protein degradation

HFF monolayers seeded in 6 cm petri dishes were infected with freshly egressed parasites and grown for 48 h in the absence or presence of IAA (500 µM). The parasites were pelleted at 1,200 rpm, resuspended in protein loading buffer containing 2% SDS, and boiled for 15 min at 95 °C. To monitor the depletion over various periods, the HFF monolayer was infected with freshly egressed parasites for 1 h at 37 °C. The extracellular parasites were then washed with DMEM supplemented with 500 µM IAA for 1, 2, 4, 8, or 12 h. After 12 h of depletion, all the samples were pelleted and resuspended in SDS. Protein depletion was assessed by WB using an anti-HA antibody against the protein of interest and anti-MIC2 as a loading control.

## Supporting information

**S1 Fig.  Additional information of the RNG2 purification from insect cells.** (**A**) SEC chromatogram of the purified RNG2-ΔCTer variant obtained via Strep-tactin affinity purification, with Coomassie-stained analysis of the protein-containing fractions. (**B**) Western blot analysis of the purification of the ΔN+CTer and ΔCTer variants. The data underlying this figure can be found in S2 Data. (**C**) Mass spectrometry analysis identified peptides belonging to the purified RNG2ΔN+CTer and ΔCTer samples. (**D**) Overlay of chromatogram traces of four different RNG2 purifications: final sample of the RNG2 ΔN+CTer construct and isolated fragments of 60, 90 and 155 kDa fragment. Elution volume of reference proteins with defined MW is shown at the bottom (orange line). The data underlying this figure can be found in S2 Data.
(TIFF)

**S2 Fig.  Multiple epitopes-tagging across RNG2. (A) Upper panel:** schematic representation of the modified endogenous locus. **Lower panel:** polymerase chain reaction of genomic extracted DNA from *Toxoplasma gondii* tachyzoites to verify the insertion of the N-terminal Myc tag/Internal Ty-Tag and C-terminal mAID-HA. (**B**) Immunofluorescence images of the RNG2-mAID protein in the absence of auxin (12 h) detected via Myc/Ty/HA. Scale bar = 3 µm.
(TIFF)

**S3 Fig.  Additional information on the processed forms and localization of the RNG2 protein. (A)** Mapping of the processed formed of RNG2 observed across a variety of western blot in RNG2 variants expressed in insect cell and the endogenous RNG2 in *Toxoplasma gondii*. Five main cleavage sites are identified shared between Sf9 cells and *T. gondii*. (**B**) Western blot analysis of a solubility assay revealed using the different tags inserted into the RNG2-mAID strain. Catalase is a solubility control soluble in PBS and IMC1 is an insoluble control soluble only in 1% SDS. (**C**) Colocalization of the Ty-Tag and αRNG2 antibody with the N-terminal (Myc) or C-terminal (HA) tags performed by U-ExM with extruded conoid. Scale bar = 3 µm. Plot profile analysis of the signal. The data underlying this figure can be found in S2 Data. (**D**) Colocalization of the Ty-Tag with the N-terminal (Myc) or C-terminal (HA) tags performed by U-ExM with retracted conoid. Scale bar = 3 µm. Plot profile analysis of the signal. The data underlying this figure can be found in S2 Data.
(TIFF)

**S4 Fig.  Detachment of the conoid in absence of RNG2 impairs parasite motility and invasion, additional phenotyping of the RNG2-mAID strain. (A)** Left panel, images of plaque assays of the RNG2-mAID strain/

Δ*rng2* strain and TIR1 in the presence or absence of auxin (IAA). The data underlying this figure can be found in S2 Data. **(B)** Two additional replicates of ethanol-triggered microneme secretion assessed by western blot. ESA = Excreted secreted antigens. **(C)** Quantification of the egress of the RNG2-mAID strain in the presence or absence of auxin (IAA). One hundred parasites were counted per replicate, and 3 biological replicates were performed. The data underlying this figure can be found in S2 Data. **(D)** Gliding trail assay of RNG2-mAID; gliding motility was determined via visualization of the SAG1 trail after BIPPO induction. **(E)** Quantification of the invasion of the RNG2-mAID strain in the presence or absence of auxin (IAA). One hundred parasites were counted per replicate, and 3 biological replicates were performed. Statistical analysis was performed via one-way ANOVA, and $p < 0.05$ was considered significant and is plotted in the figure. The data underlying this figure can be found in S2 Data.
(TIFF)

**S5 Fig. Additional information on the cryo-electron tomography performed on RNG2-mAID strain. (A)** Tomogram slices presented in Fig 4 without any overlay **(B)** Tomogram slices with and without overlay of untreated RNG2-mAID showing the correct positioning of the elements of the apical complex. **(C)** IFA images showing the localization of the Nd6 protein in presence or absence of RNG2. **(D)** U-ExM images showing the localization of the Nd6 protein in with the conoid attached or detached from the apical pole of the parasite.
(TIFF)

**S6 Fig. Additional information on the RNG2 variants expressed in UPRT locus. (A)** IFA localization and plaque assay assessment of the first round of RNG2 variants. Cell marker: IMC1. Scale bar = 2 µm. **(B)** IFA localization and plaque assay assessment of the second round of the RNG2 variant. Cell marker: GAP45. Scale bar = 2 µm **(C)** IFA localization and plaque assay assessment of the processed forms observed in insect cells and expressed in the UPRT locus. Cell marker: GAP45. Scale bar = 2 µm.
(TIFF)

**S1 Data. Additional information on antibodies and primers used in this study.** This data table lists the antibodies used in this study for Indirect-Immunofluorescence (IFA), Ultrastructure Expansion Microscopy (U-ExM), iterative Ultrastructure Expansion Microscopy (iU-ExM), western blot and Immuno-Electron Microscopy (IEM). This file also contains the list of primers used to generate all the strains presented in this study as well as the primers used to verify the correct genomic integration of relevant strains.
(XLSX)

**S2 Data. Source Numerical Data of this study.** This data table contains all the numerical data used in this study. Each sheet contains the numerical data for the respective panels.
(XLSX)

**S1 Raw images. Uncropped raw images for Figs 2, 3, 6, 7, S1, S3, and S4.**
(PDF)

**S1 Movie. Reconstruction of the Fig 4E panel I.**
(MP4)

**S2 Movie. Reconstruction of the Fig 4E panel II.**
(MP4)

**S3 Movie. Reconstruction of the Fig 4E panel III.**
(MP4)

## Acknowledgments

We are grateful to Paul Guichard, Virginie Hamel, and Vincent Louvel for their teaching and advice for iterative ultra-structure expansion microscopy, as well as Souradip Mukherjee for his technical assistance in the Cryo-ET analysis. We thank Monica E Mondragon Castelán and Sirenia González Pozos for their help in processing the IEM. Micrographs were obtained at the Electron Microscopy Facility (LaNSE, CINVESTAV-IPN, Mexico). We thank the team at the Bio-imaging Core Facility, François Prodon, Olivier Brun, and Nicolas Liaudet, for their technical assistance and imaging analysis.

## Author contributions

**Conceptualization:** Romuald Haase, Bingjian Ren, Dominique Soldati-Favre.

**Formal analysis:** Romuald Haase, Bingjian Ren, Albert Tell i Puig, Oscar Vadas.

**Funding acquisition:** Ricardo Mondragón-Flores, Dominique Soldati-Favre.

**Investigation:** Romuald Haase, Bingjian Ren, Albert Tell i Puig, Alessandro Bonavoglia, Jean-Baptiste Marq, Rémy Visentin, Nicolas Dos Santos Pacheco, Bohumil Maco, Ricardo Mondragón-Flores, Oscar Vadas.

**Methodology:** Romuald Haase, Bingjian Ren, Albert Tell i Puig, Alessandro Bonavoglia, Oscar Vadas, Dominique Soldati-Favre.

**Resources:** Ricardo Mondragón-Flores, Dominique Soldati-Favre.

**Writing – original draft:** Romuald Haase.

**Writing – review & editing:** Bingjian Ren, Albert Tell i Puig, Nicolas Dos Santos Pacheco, Ricardo Mondragón-Flores, Oscar Vadas, Dominique Soldati-Favre.

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
