## [Editor Report · Decision Letter 0]

18 Sep 2025

Dear Dr Soldati-Favre,

Thank you for submitting your Portable Peer Review manuscript entitled "RNG2 tethers the conoid to the apical polar ring in Toxoplasma gondii: a key mechanism controlling parasite motility and invasion" for consideration as a Research Article by PLOS Biology.

Your manuscript (and the PLOS Pathogens reviews and your responses) has now been evaluated by the PLOS Biology editorial staff, as well as by an academic editor with relevant expertise, and I'm writing to let you know that we would like to send your submission out for further review.

Once your full submission is complete, your paper will undergo a series of checks in preparation for peer review. After your manuscript has passed the checks it will be sent out for re-review. To provide the metadata for your submission, please Login to Editorial Manager (https://www.editorialmanager.com/pbiology) within two working days, i.e. by Sep 22 2025 11:59PM.

Kind regards,

Roli Roberts

Roland G Roberts PhD

Senior Editor

PLOS Biology

rroberts@plos.org

on behalf of

Melissa Vazquez Hernandez, Ph.D.

Associate Editor

PLOS Biology

---

## [Decision Letter · Decision Letter 1]

20 Oct 2025

Dear Dr Soldati-Favre,

Thank you for your patience while we considered your revised manuscript "RNG2 tethers the conoid to the apical polar ring in Toxoplasma gondii: a key mechanism controlling parasite motility and invasion" for publication as a Research Article at PLOS Biology. I'm handling your paper temporarily while my colleague Dr Vazquez-Hernandez is out of the office. This revised version of your manuscript has been evaluated by the PLOS Biology editors, the Academic Editor, and one of the original reviewers.

Based on the review (see the foot of this email) and on our Academic Editor's assessment of your revision, we are likely to accept this manuscript for publication, provided you satisfactorily address the following data and other policy-related requests.

a) We like to avoid punctuation in Titles. Please change it to: "RNG2 tethers the conoid to the apical polar ring in Toxoplasma gondii to enable parasite motility and invasion"

b) Please supply the original uncropped blots or gels for the appropriate Figure panels (Figs 2BCE, 3BC, 5C, 6BI, 7BJ, S1B, S2A, S3B, S4B).

c) Please address my Data Policy requests below; specifically, we need you to supply the numerical values underlying Figs 2D, 3E, 4DF, 5E, 6DEFJKL, 7DEFHL, S1AD, S3CD, S4ACE, either as a supplementary data file or as a permanent DOI’d deposition.

d) Please cite the location of the data clearly in all relevant main and supplementary Figure legends, e.g. “The data underlying this Figure can be found in S1 Data” or “The data underlying this Figure can be found in https://zenodo.org/records/XXXXXXXX

e) Please include the URLs of your funders in the Financial Disclosure statement.

We expect to receive your revised manuscript within two weeks.

*Published Peer Review History*

*Press*

Sincerely,

Roli Roberts

Roland G Roberts PhD

Senior Editor

rroberts@plos.org

on behalf of

Melissa Vazquez Hernandez, Ph.D.

Associate Editor

PLOS Biology

DATA POLICY:

Regardless of the method selected, please ensure that you provide the individual numerical values that underlie the summary data displayed in the following figure panels as they are essential for readers to assess your analysis and to reproduce it: Figs 2D, 3E, 4DF, 5E, 6DEFJKL, 7DEFHL, S1AD, S3CD, S4ACE. NOTE: the numerical data provided should include all replicates AND the way in which the plotted mean and errors were derived (it should not present only the mean/average values).

CODE POLICY

We require the original, uncropped and minimally adjusted images supporting all blot and gel results (Figs 2BCE, 3BC, 5C, 6BI, 7BJ, S1B, S2A, S3B, S4B) reported in an article's figures or Supporting Information files. We will require these files before a manuscript can be accepted so please prepare and upload them now. Please carefully read our guidelines for how to prepare and upload this data: https://journals.plos.org/plosbiology/s/figures#loc-blot-and-gel-reporting-requirements

DATA NOT SHOWN?

REVIEWER'S COMMENTS:

Reviewer #2:

The revision of the article by Hasse et al. represents a significantly improved version of the manuscript, now presenting conclusions that are rigorously supported by two sets of new experiments. In particular, the authors have abandoned their initial model, which proposed that proteolytic processing and oligomerization of RNG2 could provide the flexibility required to span the distance from the APR to the conoid. The revised conclusion—that RNG2 is sufficiently long to bridge this intervening space—does not diminish the scope or impact of the study. My second major criticism, regarding microneme secretion, has also been carefully addressed, and the authors now clarify the primary phenotype associated with RNG2. Furthermore, the inclusion of cryo-ET analysis of the RNG2 mutant adds valuable structural information on RNG2 defect in apical complex. Finally, the authors provide a well-reasoned explanation for the apparent discrepancy between the 50% defect in rhoptry secretion and the detachment of rhoptries from the apex observed in nearly all mutants.

---

## [Editor Report · Decision Letter 2]

6 Nov 2025

Dear Dominique,

Thank you for the submission of your revised Research Article "RNG2 tethers the conoid to the apical polar ring in Toxoplasma gondii to enable parasite motility and invasion" for publication in PLOS Biology. On behalf of my colleagues and the Academic Editor, Kami Kim, I am pleased to say that we can in principle accept your manuscript for publication, provided you address any remaining formatting and reporting issues. These will be detailed in an email you should receive within 2-3 business days from our colleagues in the journal operations team; no action is required from you until then. Please note that we will not be able to formally accept your manuscript and schedule it for publication until you have completed any requested changes.

*Just a quick note. I took a quick look through the manuscript and I noticed that in some figures the legends seemed to have the description letter wrong. For example on Figure 6. Just letting you know for when you have to do the checks.

PRESS

Sincerely, 

Melissa

Melissa Vazquez Hernandez, Ph.D., Ph.D.

Associate Editor

PLOS Biology
